# Variational Disentanglement for Domain Generalization

**Yufei Wang**                                                              *yufei001@ntu.edu.sg*
*Nanyang Technological University*

**Haoliang Li**                                                          *haoliang.li@cityu.edu.hk*
*City University of Hong Kong*

**Hao Cheng**                                                               *hao006@ntu.edu.sg*
*Nanyang Technological University*

**Bihan Wen**\*                                                           *bihan.wen@ntu.edu.sg*
*Nanyang Technological University*

**Lap-pui Chau**                                                      *lap-pui.chau@polyu.edu.hk*
*The Hong Kong Polytechnic University*

**Alex C. Kot**                                                             *eackot@ntu.edu.sg*
*Nanyang Technological University*

**Reviewed on OpenReview:** *https://openreview.net/forum?id=fudOtITMIZ*

## Abstract

Domain generalization aims to learn a domain-invariant model that can generalize well to the unseen target domain. In this paper, based on the assumption that there exists an invariant feature mapping, we propose an evidence upper bound of the divergence between the category-specific feature and its invariant ground-truth using variational inference. To optimize this upper bound, we further propose an efficient Variational Disentanglement Network (VDN) that is capable of disentangling the domain-specific features and category-specific features (which generalize well to the unseen samples). Besides, the generated novel images from VDN are used to further improve the generalization ability. We conduct extensive experiments to verify our method on three benchmarks, and both quantitative and qualitative results illustrate the effectiveness of our method.

## 1 Introduction

Nowadays, deep neural networks are widely used in numerous tasks and exhibit remarkable performance. However, their performance may degrade rapidly when the deployed environment is different from the training one. How to obtain a domain invariant network that can generalize to the data collected from an unseen environment is always a research hotspot (Zhang et al., 2016; Kawaguchi et al., 2017).

Domain generalization (DG) aims to tackle the generalization problem where the data from the target domain is inaccessible. Generally, existing DG research works can be categorized into three streams, invariant feature representation learning, meta-learning, and data augmentation. The objective of invariant feature representation learning is to extract shareable information across different domains, as such, the learned model is expected to be better generalized to the unseen but related domain during evaluation. For example, Li et al. (2018b;d) aim to minimize the divergence of the latent features between different domains or the divergence with a pre-defined prior distribution. Dou et al. (2019) further propose to minimize the divergence of the distributions conditioned on the category label. Although some desired performances have been reported through the aforementioned methods, they may face the risk of overfitting to the source domains without carefully specifying the invariant information to learn, e.g., barely ensuring invariance may cause over-matching on the observed domains (Shui et al., 2022). On the other hand, data augmentation based

---

\*Corresponding author.

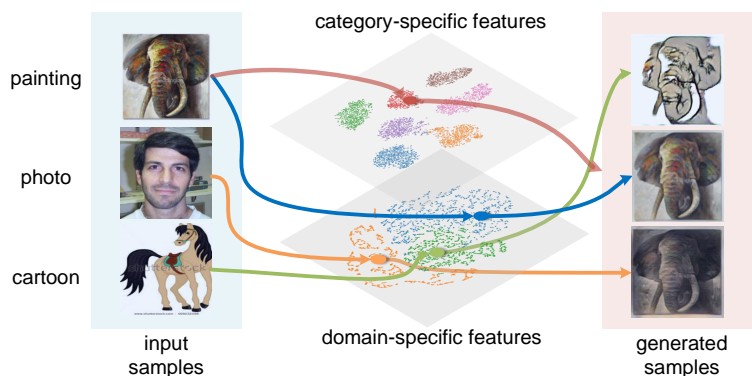

Figure 1: Given data from multiple source domains, we propose to learn a domain invariant embedding for classification and a domain-specific embedding to encode the style information. The arrows from images to embeddings indicate the encoding process, and the arrows from features to the generated samples represent the generation process. Invariant category-specific features are learned using the proposed variational disentanglement framework.

methods have also been proved to be effective in the domain generalization setting by enlarging the scale of training data with different augmentation strategies, e.g., data augmentation through adversarial training (Volpi et al., 2018; Zhou et al., 2020b), MixUp (Wang et al., 2020c), stacked transformation (Zhang et al., 2020), and Fourier transformation (Xu et al., 2021). However, these augmentation strategies can be limited since they only focus on some specific types of data augmentation, as such, the diversity of augmented data may be constrained.

In this paper, we propose to learn an invariant feature representation by matching the distribution of the feature space across domains to a distribution which is expected to represent the ground truth (i.e., invariant information) for the problem of domain generalization on a specific task. We first propose to derive the evidence upper bound of the divergence between the distribution of the feature space across domains and its unknown ground-truth distribution. To further optimize the upper bound, we develop an effective framework named Variational Disentanglement Network (VDN), which is capable of alleviating the aforementioned limitations jointly. More specifically, instead of optimizing the latent feature $z$ alone, we propose to optimize the category(label)-specific feature $z_c$ and domain-specific feature $z_d$ at the same time. The optimization of $z_c$ and $z_d$ are twofold: minimizing the redundant information in the task-specific features and maximizing the posterior probability of the generated samples with random styles. By maximizing the posterior probability term, $z_c$ and $z_d$ are impelled to be disentangled and contain all necessary information to avoid a trivial solution that overfits to source domains. Meanwhile, minimizing our proposed information term that filters out redundant information in $z_c$ can be regarded as the strategy of information bottleneck which benefits the generalization ability of the model (Alemi et al., 2016). Last but not the least, our VDN is also capable of generating different and diverse data across domains based on the specific task by swapping $z_d$ across domains. The generated novel images can be further used to improve the generalization ability of the model, and are also compatible with existing augmentation-based methods. Extensive experiments are conducted on widely-used benchmarks, and both quantitative and qualitative results show the effectiveness of our method.

In summary, our contributions are as follows:

- We provide a theoretical insight of how variational disentanglement can benefit the task of domain generalization based on the basic assumption of the invariant feature representation learning.

- We propose to derive a novel evidence upper bound of the divergence between the distribution of task-specific features and its invariant ground truth for domain generalization. Our proposed evidence upper bound further supports the rationality and build the tie between previous feature alignment-based methods (Li et al., 2018b)[1] and disentangle-based methods, e.g., Li et al. (2017); Peng et al. (2019b).

- A novel framework is proposed to optimize the proposed evidence upper bound of the divergence. Extensive experiments demonstrate the effectiveness and superiority of our proposed method.

---

[1]In Li et al. (2018b), the author empirically proposes to use an adversarial autoencoder to minimize the divergence between $p_h$ and $p_x$ where $p_h$ is the latent code distribution and $p_x$ is the prior distribution (Laplace distribution in the paper).

## 2 Related works

### 2.1 Domain generalization

The core ideas of domain generalization (DG) are inherited from domain adaptation (Huang et al., 2006; Pan et al., 2011; Zhang et al., 2015; Ghifary et al., 2017; Blanchard et al., 2021) to some extent, e.g., they assume that there exists something in common between different domains even if they can look quite different. For example, Khosla et al. (2012); Seo et al. (2020) aim to jointly learn an unbiased model and a set of bias vectors for each domain, Yang & Gao (2013) use Canonical Correlation Analysis (CCA) to extract the shared feature, Muandet et al. (2013) propose a domain invariant analysis method which used MMD and was further extended by Li et al. (2018b), multi-task autoencoders were also used by Ghifary et al. (2015) to learn a shared feature extractor with multiple decoders. Moreover, various regularization methods of latent code are proposed (Zhao et al., 2020; Wang et al., 2020b), e.g., low-rank regularization (Xu et al., 2014; Li et al., 2017; 2020; Piratla et al., 2020). In addition, data augmentation based methods are also proved to be effective in the domain generalization setting, e.g., GAN generated samples (Volpi et al., 2018; Zhou et al., 2020b), domain mixup (Wang et al., 2020c; Zhou et al., 2021a), stacked transformations (Zhang et al., 2020), domain-guided perturbation direction (Shankar et al., 2018), Fourier augmentation (Xu et al., 2021), and solving a jigsaw puzzle problem (Carlucci et al., 2019). Meta-learning based methods (Li et al., 2018a; Balaji et al., 2018; Li et al., 2019b;a; Dou et al., 2019; Du et al., 2020) are also explored by learning from episodes that simulate the domain gaps. Recently, invariant risk minimization (IRM) is proposed to eliminate spurious correlations as such the models are expected to have better generalization performance (Arjovsky et al., 2019; Ahuja et al., 2020; Bellot & van der Schaar, 2020; Zunino et al., 2020). As for feature disentanglement, current methods are usually based on decomposition (Li et al., 2017; Khosla et al., 2012; Piratla et al., 2020; Chattopadhyay et al., 2020). Generation based disentanglement methods are also explored. For instance, Peng et al. (2019b) conduct the single domain generalization using adversarial disentangled auto-encoder and Wang et al. (2020a) provide a pair of encoders for disentangled features in each domain.

### 2.2 Feature disentanglement and image translation

Our proposed method is related to the feature disentanglement, which has also been widely adopted in the problem of cross-domain learning (Bousmalis et al., 2017; Hoffman et al., 2018; Russo et al., 2018; Saito et al., 2018). A lot of progress in domain generalization has been made by applying feature disentanglement (Peng et al., 2019b; Khosla et al., 2012; Li et al., 2017; Piratla et al., 2020). For instance, Peng et al. (2019b) propose a domain agnostic learning method based on VAE and adversarial learning, and Khosla et al. (2012); Li et al. (2017); Piratla et al. (2020) assume that there exists a shared model which is regarded as domain invariant and a set of domain-specific weights.

Our work is also related to the image translation (Isola et al., 2017; Choi et al., 2018; Zhu et al., 2017; Liu et al., 2019), which can be treated as conducting feature disentanglement by separating the style feature and content feature. Generally, the existing translation methods can be categorized into two streams, supervised/paired (Isola et al., 2017) and unsupervised/unpaired (Zhu et al., 2017; Choi et al., 2018). While some progress has been achieved, most of the existing models require a large amount of data. Recently, some efforts have been made to improve the capability of the translation model by utilizing few samples (Liu et al., 2019; Saito et al., 2020). Specifically, the framework we propose conducts translation tasks in a more efficient manner given limited and diverse data. This helps us obtain a more accurate posterior probability estimation. In addition, the generated high-quality samples with different combinations of image attributes (e.g., samples with a new combination of angles, shape, and color (Zhao et al., 2018)) based on our proposed framework are also used for data augmentation purpose for better generalization capability.

## 3 Methodology

### 3.1 Preliminary

Assume we observe $K$ domains and there are $N_i$ labeled samples $\{(x_j, y_j)\}_{j=1}^{N_i}$ in the domain $\mathbf{D}_i$. $X_i$ is the image set of the domain $D_i$ and $Y$ is the set of labels. We first introduce the assumption regarding the invariant feature representation for the task of DG.

**Assumption 1** *(Learnability) Denote the dimension of the image $x$ and the latent feature $z_c$ by $d_x$ and $d_c$, respectively, let $\phi_c : \mathbb{R}^{d_x} \to L^1(\mathbb{R}^{d_c})$ be the mapping between an image $x$ and its probability density function (PDF) $f(z_c|x)$, where*

$L^1$ denotes 1-norm integrable function space. A deterministic mapping $\phi_g : \mathbb{R}^{d_c} \to \mathbb{N}$ acts as a classifier to predict the category of the image based on its latent feature $z_c$. For a domain generalization problem with $n_s$ source domains ($X_s = X_1 \cup X_2 ... \cup X_{n_s}$) and a target domain $X_t$, we say it learnable if

$$
\begin{aligned}
&\exists \phi_c, \quad s.t. \\
&\mathbb{E}_{x \sim P_{x_{i|y}}}[\phi_c(x)] = \mathbb{E}_{x \sim P_{x_{j|y}}}[\phi_c(x)], \forall y \in Y, \forall i, j \in \{1, 2, ..., n_s, t\} \\
&\phi_g(\psi(\phi_c(x))) = y, \forall x \in X_s \cup X_t,
\end{aligned}
\tag{1}
$$

where $P_{x_{i|y}}$ represents the conditional distribution of images with the category $y$ in the domain $i$, and $\psi(\phi_c(x)) = E_{z_c \sim \phi_c(x)} z_c$.

While we assume a noiseless case similar to Tachet des Combes et al. (2020), such an assumption is mild and is commonly adopted in the community of domain generalization (Muandet et al., 2013). Based on the assumption that if the task is feasible, there should exist a domain invariant mapping $\phi_c(\cdot)$. In other words, we can improve the generalization capability of the classifier model if we can find the domain-invariant mapping $\phi_c$. However, as suggested in Kingma & Welling (2013), the invariant ground-truth distribution $P(\mathbf{z_c}|x)$ (where $P(\mathbf{z_c}|x) = \phi_c(x)$) can be intractable, as such, we are unable to obtain an explicit form of $P(\mathbf{z_c}|x)$. To this end, we propose to use variational inference to find an approximation of $\phi_c(\cdot)$. [2]

## 3.2 Motivation

To approximate the conditional distribution of domain-invariant features $\phi_c$, we propose to minimize the evidence upper bound of the KL divergence $\mathcal{D}(Q(\mathbf{z_c}|x)||P(\mathbf{z_c}|x))$[3] derived by Bayes variational inference, where $Q(\mathbf{z_c}|x)$ denotes the feature distribution obtained from the category-specific encoder $E_c$ and $P(\mathbf{z_c}|x)$ is the invariant ground truth distribution. Since we aim to find a deterministic classifier, we set $E_c(x) = \psi(Q(\mathbf{z_c}|x))$ for simplicity. We now introduce the rationality of learning invariant feature representation by minimizing the upper bound of KL divergence. Due to the limited space, proofs of Theorem 1 and 2, and additional details are placed in the Appendix.

**Lemma 1** *The KL divergence $\mathcal{D}(Q(\mathbf{z_c}|x)||P(\mathbf{z_c}|x))$ can be represented as*

$$
\mathcal{D}(Q(\mathbf{z_c}|x)||P(\mathbf{z_c})) - E_{z_c \sim Q_{z_c|x}}[\log p(x|z_c)] + \log p(x),
$$

where $p(x|z_c)$ is the PDF of the distribution $P(\mathbf{x}|z_c)$. Based on Lemma 1, we can derive the evidence upper bound of $\mathcal{D}(Q(\mathbf{z_c}|x)||P(\mathbf{z_c}|x))$.

**Theorem 1** *The evidence upper bound of KL divergence $\mathcal{D}(Q(\mathbf{z_c}|x)||P(\mathbf{z_c}|x))$ between the distribution $Q(\mathbf{z_c}|x)$ and the ground-truth $P(\mathbf{z_c}|x)$ is as follows:*

$$
\underbrace{\mathcal{D}(Q(\mathbf{z_c}|x)||P(\mathbf{z_c}))}_{\text{① information gain term}} - \underbrace{E_{z_c \sim Q_{z_c|x}, z_d \sim P_{z_d}}[\log p(x|z)]}_{\text{② posterior probability term}} + C,
$$

where $C$ is a constant, $z = [z_c, z_d]$ and $P_d$ can be an arbitrary prior distribution.

It is worth noting that $z_c$ and $z_d$ are implicitly determined by the category and domain label respectively, i.e., only the realistic image in accord with its corresponding category and domain could have a high probability density value. We further show that the performance in the unseen target domain can also benefited from the optimization of the given upper bound in source domains.

---

[2]Similar to other DG works (Li et al., 2018d;c; Hu et al., 2020), there is an implicit assumption that the label prior does not vary a lot among source and target domains. The cases that label priors are largely different may in the scope of imbalanced classification (Sun et al., 2009) and heterogeneous domain generalization (Wang et al., 2020c; Li et al., 2019b). In addition, theoretical analysis can be found in Shui et al. (2021) which demonstrates that the upper bound of target risk is related to the discrepancy of label distribution.

[3]For consistency, we represent distributions using upper case letter, e.g., $Q(\mathbf{z_c}|x)$ means the distribution of random variable $\mathbf{z_c}$ conditioned on $x$ and it is also abbreviated as $Q_{z_c|x}$. The lower case represent the exact probability density value, e.g., $p(x|z_c, z_d)$.

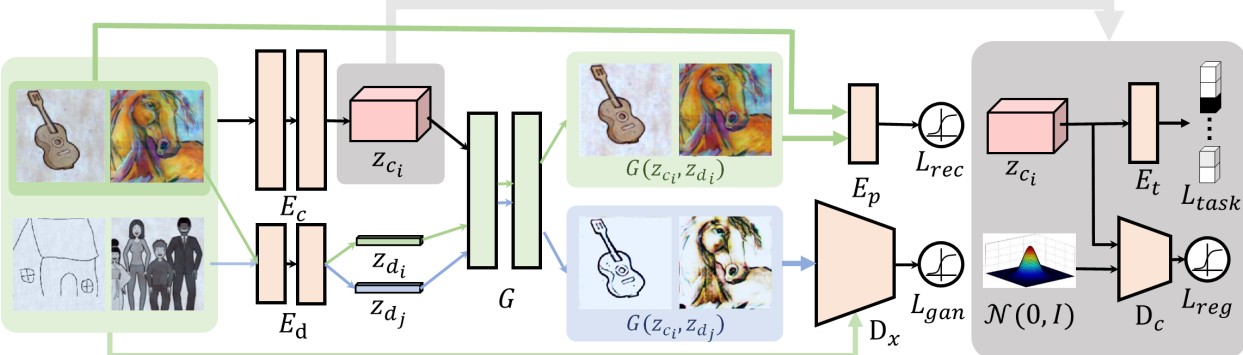

Figure 2: Framework of our method for training. For a random sample $(x_i, y_i)$ in the training set, we random select another sample $(x_j, y_j)$ that does not require the same category. We feed the two samples into the task-specific encoder $E_c$ and domain-specific encoder $E_d$ and get the corresponding code $z_{c_i}, z_{c_j}$ and $z_{d_i}, z_{d_j}$. Then we use the generator $G$ to obtain the reconstructed samples $G(z_{c_i}, z_{d_i})$ and samples with random style $G(z_{c_i}, z_{d_j})$. The perceptual loss and discriminator loss are used to enhance the quality of generated samples which further guarantee good disentanglement between class-specific feature $z_c$ and domain-specific feature $z_d$. In addition, for the task-specific feature $z_c$, it is supervised by the specific task, e.g., classification, and regularized by the information gain term $\mathcal{D}(Q(\mathbf{z_c}|x)||P(\mathbf{z_c}))$.

**Assumption 2** *Given a sample $x^t$ from the target domain $X_t$, there exists a non-empty feasible set $\mathcal{I}$ which is defined as*

$$\mathcal{I} = \{I | q(z_c|x^t) \le \sum_{i \in I} \beta_i q(z_c|x_i^s), \forall z_c \in \mathbb{R}^{d_c}\} \cap \{I | \phi_c(x^t) = \phi_c(x_i^s), \forall i \in I\},$$

*where $I$ is the index set, $x_i^s$ denotes an arbitrary sample with index $i$ in any source domains, and $q(z_c|x)$ is the probability density function value of $z_c$ conditioned on $x^t$ from distribution $Q(\mathbf{z_c}|x^t)$.*

Assumption 2 is a mild assumption since it holds as long as the task is feasible (the second set is not empty) and $q(z_c|x)$ is a distribution that satisfies $q(z_c|x) > 0$ given any $z_c$, e.g., Gaussian distribution (the first set is not empty).

**Theorem 2** *Based on Assumption 2, the KL divergence between $Q(\mathbf{z_c}|x^t)$ and the unknown domain-invariant ground truth distribution $P(\mathbf{z_c}|x^t)$ can be bounded as follows*

$$\mathcal{D}(Q(\mathbf{z_c}|x^t)||P(\mathbf{z_c}|x^t)) \le \inf_{I \in \mathcal{I}} [\sum_{i \in I} \beta_i \mathcal{D}(Q(\mathbf{z_c}|x_i^s)||P(\mathbf{z_c}|x_i^s))].$$

The above theorem demonstrates that the KL divergence between $Q(\mathbf{z_c}|x)$ and $P(\mathbf{z_c}|x)$ from source domains constitutes the divergence upper bound in the unseen target domain. Therefore, it further supports the rationale and effectiveness of our method.

### 3.3 Optimization strategies

In this section, we introduce the overall framework and the optimization details of each term in our proposed evidence upper bound.

#### 3.3.1 Overall framework

The whole framework of our proposed method is illustrated in Fig. 2 which consists of a task-specific encoder $E_c$, a domain-specific encoder $E_d$, a generator $G$, a discriminator $D_x$ to distinguish real and generated images, a discriminator $D_c$ to justify whether the task-specific feature comes from a predefined distribution, and a task-specific network $E_t$ for classification purpose. The overall optimization objective is given as

$$L = \underbrace{\lambda_{reg} L_{reg}}_{\text{To minimize } ①} + \underbrace{L_{posterior}}_{\text{To minimize } ②}, \quad (2)$$

where the first term $L_{reg}$ is defined as follows

$$L_{reg} = \mathbb{E}_{z_c \sim P_{z_c}}[D_c(z_c)] + \mathbb{E}_{x \sim X}[\log(-D_c(E_c(x)))] + 1, \tag{3}$$

which acts as a regularization term that aligns the task-specific feature to a predefined distribution and is further interpreted in section 3.3.2, and $L_{posterior}$ is defined as

$$L_{posterior} = L_{task} + \lambda_{rec}L_{rec} + \lambda_{gan}L_{gan}, \tag{4}$$

where the definitions of each term in $L_{posterior}$ are defined as follows

$$L_{task} = E_{x \sim X}[l_{task}(E_t \circ E_c(x), y)] + E_{z_c \sim Q_c, z_d \sim Q_d}[l_{task}(E_t \circ E_c(G(z_c, z_d)), y)],$$
$$L_{gan} = E_{x \sim X}[D_x(x|z_c, z_d)] - E_{z_c \sim Q_c, z_d \sim Q_d}[D_x(G(z_c, z_d)|z_c, z_d)],$$
$$L_{rec} = E_{x \sim X}||E_p(x) - E_p(G(E_c(x), E_d(x)))||_1.$$

The details of each term in $L_{posterior}$ are illustrated in section 3.3.3. These losses are optimized jointly to disentangle the task-specific feature $z_c$ and domain-specific feature $z_d$.

In the test phase, an image from an unseen but related target domain is passed through the task-specific encoder $E_c$ and the task-specific network $E_t$, and the other networks will not be involved. Therefore, our network is supposed to have the same inference complexity as the vanilla model.

### 3.3.2 Estimation of the information gain term

To reduce the risk of learned task-specific feature overfitting to the source domains, similar to Li et al. (2018b), we minimize the divergence between $Q(\mathbf{z_c}|x)$ which is introduced by the task-specific encoder $E_c$, and a predefined prior distribution $P(\mathbf{z_c})$. The divergence $\mathcal{D}(Q(\mathbf{z_c}|x)||P(\mathbf{z_c}))$ can be interpreted as the information gain by introducing a specific image $x$. Ideally, the optimal features only contain the necessary information for classification, and domain-specific information will be ruled out. To this end, we minimize the task-specific loss, i.e., classification loss in equation 12, and information gain term in our framework simultaneously. For the minimization of information gain term, we empirically find that directly optimizing this term using the reparameterization trick (Kingma & Welling, 2013) may not be feasible due to the high dimensionality of $z_c$. To this end, we propose to optimize $\mathcal{D}(Q(\mathbf{z_c})||P(\mathbf{z_c}))$, which is equivalent to optimize the upper bound of the information gain term $L_{reg}$ in Theorem 1. The proof is in Lemma A.2 and the optimization of the proposed upper bound also supports the core idea of the previous work (Li et al., 2018b) that aligns the latent feature to a pre-defined distribution. More specifically, we optimize its dual form following the core idea of f-GAN (Nowozin et al., 2016) given as

$$\mathcal{D}(Q(\mathbf{z_c})||P(\mathbf{z_c})) = D_f(P(\mathbf{z_c})||Q(\mathbf{z_c})) = \int q(z_c) \sup_{t \in \text{dom}_{f^*}} \left\{ t \frac{p(z_c)}{q(z_c)} - f^*(t) \right\} dz_c, \tag{5}$$

where $p(\cdot)$ and $q(\cdot)$ are PDFs of $P$ and $Q$ respectively, $f(u) = -\log(u)$, and $f^*(t)$ denotes its Fenchel conjugate (Hiriart-Urruty & Lemaréchal, 2004) which is defined as

$$f^*(t) = \sup_{u \in \text{dom}_f} \{ut - f(u)\} = -1 - \log(-t). \tag{6}$$

One can replace $t$ in equation 6 using an arbitrary class of functions $T$, as such, the term ① can be represented as

$$\mathcal{D}(Q(\mathbf{z_c})||P(\mathbf{z_c})) = \sup_{T \in \mathcal{T}} \left( \mathbb{E}_{z_c \sim P_{z_c}}[T(z_c)] - \mathbb{E}_{z_c \sim Q_{z_c|x}}[f^*(T(z_c))] \right), \tag{7}$$

if the capacity of $\mathcal{T}$ is large enough.

To optimize equation 7, an adversarial training manner is used. Specifically, we optimize the task-specific encoder $E_c$ to maximize this term and the discriminator $D_c$ to minimize it. The final goal of the training for the regularization term is as follows

$$\min_{E_c} \max_{D_c} L_{reg} = \mathbb{E}_{z_c \sim P_{z_c}}[D_c(z_c)] + \mathbb{E}_{x \sim X}[\log(-D_c(E_c(x)))] + 1, \tag{8}$$

where the function $T$ in equation 7 is implemented by a deep neural network $D_c$ with the activation function $g_f(v) = -\exp(-v)$ at the output to retain the original domain $\text{dom}_{f^*}$.

---

**Algorithm 1** Variational Disentanglement for domain generalization.

---

**Input:** $X = \{x_i\}_{i=1}^N$, $Y = \{y_i\}_{i=1}^N$, initialized parameters $G$, $D_c$, $D_x$, $E_c$, $E_d$ and $E_t$.
**Output:** Learned parameters $G^*$, $D_c^*$, $D_x^*$, $E_c^*$, $E_d^*$ and $E_t^*$.

1:  **while** Stopping criterion is not met **do**
2:      Sample a *minibatch* $\tilde{X}$ and $\tilde{Y}$ from $X$ and $Y$ respectively.
3:      Calculate the task-specific feature $Z_c = E_c(\tilde{X})$ and domain-specific feature $Z_d = E_d(\tilde{X})$.
4:      Shuffle the domain-specific feature $\hat{Z}_d = \text{shuffle}(Z_d)$
5:      Generate the reconstructed samples $X_{rec} = G(Z_c, Z_d)$ and the samples with the random style $X_{rand} = G(Z_c, \hat{Z}_d)$.
6:      Calculate the weighted loss $L_{gen} = L_{task} + \lambda_{reg}L_{reg} + \lambda_{gan}L_{gan} + \lambda_{rec}L_{rec}$
7:      Compute the gradient of $L_{gen}$ w.r.t $G$, $E_c$, $E_d$ and $E_t$ and then do the update.
8:      **if** Update frequency is met **then**
9:          Calculate the weighted loss $L_{dis} = -(\lambda_{reg}L_{reg} + \lambda_{gan}L_{gan})$
10:         Compute the gradient of $L_{dis}$ w.r.t $D_c$ and $D_x$ and then do the update.
11:     **end if**
12: **end while**

---

### 3.3.3   Estimation of the posterior probability term

For better disentanglement, we propose to maximize the posterior probability term $p(x|z_c, z_d)$. The advantages of maximizing this term can be roughly summarized into two points: first, directly minimizing the information gain term and task-specific term may cause the overfitting to the training samples. For instance, when the dataset is not large enough, directly memorizing all the datasets may have less information gain comparing with extracting discriminate features for classification. By maximizing the posterior probability, we can guarantee that $z_c$ and $z_d$ together contain almost all the information of the image which avoid the loss of discriminative features caused by minimizing the information gain term. Second, by improving the quality of generated samples using random combined task-specific feature $z_c$ and style feature $z_d$, the task-specific and domain-specific features will be disentangled since our discriminator can not only distinguish real and fake images but can also differentiate whether the generated samples are from the specific domains thanks to the domain labels.

To maximize the posterior probability term, two obstacles need to be solved. First, an efficient sampling strategy is needed on account that the space of $z_c$ and $z_d$ is large and intractable and it is not feasible to sample ergodically from them. To this end, inspired by VAE (Kingma & Welling, 2013), we propose to adopt the task-specific encoder $E_c$ and domain-specific encoder $E_d$ instead, i.e., $Q_{z_c} \sim E_c(X)$ and $Q_{z_d} \sim E_d(X)$, to conduct code sampling of $z_c$ and $z_d$ which are most likely to generate a realistic sample. To sample in $Q_c$ and $Q_d$ independently, we shuffle the domain-specific features in a batch and ensure the generated samples $G(z_{c_i}, z_{d_j})$ using randomly combined features as realistic as possible. More details can be found in Algorithm 1.

Unlike VAE (Kingma & Welling, 2013) which only computes the reconstruction term $p(x|z_{c_i}, z_{d_i})$, we also need to compute the probability of the generated one $G(z_{c_i}, z_{d_j})$ using the generator $G$ for $\forall i \neq j$ without a corresponding ground-truth. To this end, we estimate the probability $p(x|z_c, z_d)$ using the following equations

$$p(x|z_c, z_d) = \begin{cases} La(x|G(z_c, z_d), \beta)) & \text{w/ ground-truth} \\ D_x(G(z_c, z_d)|z_c, z_d) & \text{w/o ground-truth} \end{cases}, \tag{9}$$

where $La$ denotes the PDF of Laplace distribution (corresponding to the L1 norm) to measure the pixel reconstruction loss, and $D_x(\cdot|z_c, z_d)$ is the estimation from the discriminator based on the corresponding category and domain labels/features. To optimize the term in equation 9, $D_x$ needs to have the capability to distinguish between the real and generated images $G(z_{c_i}, z_{d_j})$ through random combined latent features, and to differentiate whether the samples have the desired domain and category. Meanwhile, the generator $G$ is required to produce realistic outputs. To this end, we train the model in an adversarial manner given below

$$\min_{E_c, E_d, G} \max_{D_x} L_{gan} = E_{x \sim X}[D_x(x|z_c, z_d)] - E_{z_c \sim Q_c, z_d \sim Q_d}[D_x(G(z_c, z_d)|z_c, z_d)]. \tag{10}$$

For the reconstructed images with ground-truth, we empirically find that directly minimizing the pixel level divergence can lead to a performance degradation. To this end, we minimize the divergence of semantic features through a pre-

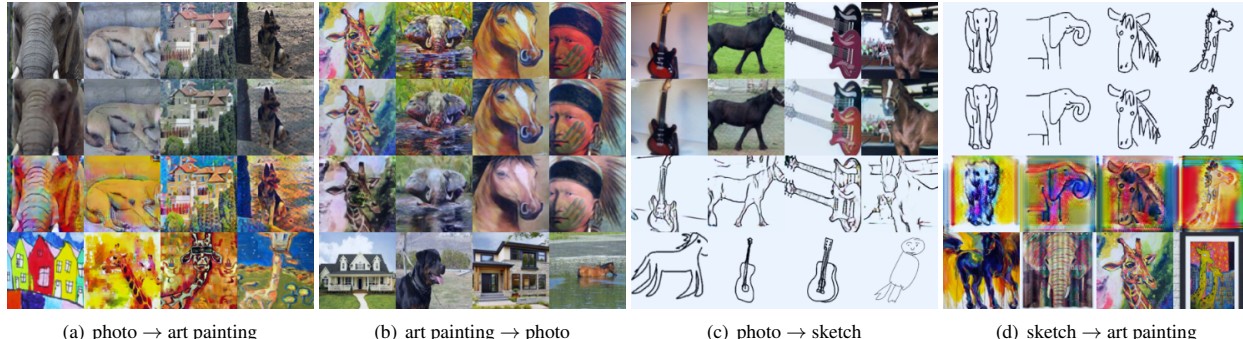

| (a) photo → art painting | (b) art painting → photo | (c) photo → sketch | (d) sketch → art painting |

Figure 3: The generated samples in source domains. The images in the first row are the input images that we want to keep the category information and in the last row are the input images that provide the style information. The second row is the reconstructed one and the third row is the translated one.

trained perceptual network $E_p$ instead. Therefore, we maximize the improved version $La(E_p(x)|E_p(G(z_c, z_d)), \beta))$ by minimizing its corresponding L1 loss given as

$$\min_{E_c, E_d, G} L_{rec} = E_{x \sim X} ||E_p(x) - E_p(G(E_c(x), E_d(x)))||_1. \tag{11}$$

Last but not least, to further enforce the task-specific encoder $E_c$ to generate more meaningful embeddings, we use label information to guide the model training by minimizing the following equation

$$\min_{E_c, E_t} L_{task} = E_{x \sim X}[l_{task}(E_t \circ E_c(x), y)] + E_{z_c \sim Q_c, z_d \sim Q_d}[l_{task}(E_t \circ E_c(G(z_c, z_d)), y)], \tag{12}$$

where $y$ denotes the ground-truth label of the input $x$ or its task-specific feature $z_c$, $l_{task}$ is the task-specific loss, e.g., cross entropy is used in our work for classification tasks. Besides, we also consider the generated samples for data augmentation purpose, which is the second term of equation 12.

## 4 Experiments

We evaluate our method using three benchmarks including Digits-DG (Zhou et al., 2020b), PACS (Li et al., 2017) and mini-DomainNet (Zhou et al., 2020c; Peng et al., 2019a). For the hyper-parameters, we set $\lambda_{reg}$, $\lambda_{rec}$ and $\lambda_{gan}$ as 0.1, 1.0 and 1.0, respectively, for all experiments. The network architectures and training details are illustrated in the supplementary materials. We report the accuracy of the model at the end of the training.

### 4.1 Digits-DG

**Settings:** Digits-DG (Zhou et al., 2020b) is a benchmark composed of MNIST, MNIST-M, SVHN, and SYN where the font and background of the samples are diverse. Except that we additionally use the Fourier-based data augmentation (Xu et al., 2021), we follow the experiment setting in Zhou et al. (2020b). More specifically, the input images are resized to 32×32 based on RGB channels. The ConvNet is used as the backbone for all methods and is divided into two parts: the task-specific encoder $E_c$ which includes the first three conv blocks except for the last max-pooling layer and the rest are regarded as the task-specific network $E_t$. For the $E_c$ and $E_t$, we use the same learning parameters with Zhou et al. (2020b). SGD is used as the optimizer with an initial learning rate of 0.05 and weight decay of 5e-5. For the rest of the networks, we use RmsProp without momentum as the optimizer with the initial learning rate of 5e-5 and the same weight decay. We train the model for 60 epochs using the batch size of 128 and all the learning rate is decayed by 0.1 at the 50th epoch.

**Results:** We repeat the experiment 3 times and report the average accuracy in Table 1. Our method has about 10% percent improvements in MNIST-M, SVHN, and SYN domains compared with the DeepAll method. In addition, the results demonstrate that our method achieves the best overall performance, especially in the domain of MNIST-M with about 3% improvement compared with other competitors.

Table 1: Evaluation of DG on the Digits-DG benchmark. The average target domain accuracy are reported.

| Method | Reference | MNIST | MNIST-M | SVHN | SYN | Avg. |
|--------|-----------|-------|---------|------|-----|------|
| DeepAll | - | 95.8 | 58.8 | 61.7 | 78.6 | 73.7 |
| MMD-AAE (Li et al., 2018a) | CVPR 2018 | 96.5 | 58.4 | 65.0 | 78.4 | 74.6 |
| CrossGrad (Shankar et al., 2018) | ICLR 2018 | 96.7 | 61.1 | 65.3 | 80.2 | 75.8 |
| DDAIG (Zhou et al., 2020b) | AAAI 2020 | 96.6 | 64.1 | 68.6 | 81.0 | 77.6 |
| L2A-OT (Zhou et al., 2020a) | ECCV 2020 | 96.7 | 63.9 | 68.6 | 83.2 | 78.1 |
| MixStyle (Zhou et al., 2021a) | ICLR 2021 | 96.5 | 63.5 | 64.7 | 81.2 | 76.5 |
| FACT (Xu et al., 2021) | CVPR 2021 | **97.9** | 65.6 | 72.4 | **90.3** | 81.5 |
| VDN | Ours | 97.6 | **68.1** | **72.9** | 87.6 | **81.6** |

Table 2: Evaluation of DG on the PACS benchmark. The average target domain accuracy of five repeated experiments is reported.

| Method | Reference | Art | Cartoon | Photo | Sketch | Avg. |
|--------|-----------|-----|---------|-------|--------|------|
| DeepAll | - | 77.0 | 75.9 | 95.5 | 70.3 | 79.5 |
| MASF (Dou et al., 2019) | NIPS 2019 | 80.3 | 77.2 | 93.9 | 71.7 | 81.0 |
| Epi-FCR (Li et al., 2019a) | ICCV 2019 | 82.1 | 77.0 | 93.9 | 73.0 | 81.5 |
| L2A-OT (Zhou et al., 2020a) | ECCV 2020 | 83.3 | 78.2 | **96.2** | 73.6 | 82.8 |
| RSC (Huang et al., 2020) | ECCV 2020 | 78.9 | 76.9 | 94.1 | 76.8 | 81.7 |
| MixStyle (Zhou et al., 2021a) | ICLR 2021 | 84.1 | 78.8 | 96.1 | 75.9 | 83.7 |
| DAML (Shu et al., 2021) | CVPR 2021 | 83.0 | 74.1 | 95.6 | 78.1 | 82.7 |
| FACT (Xu et al., 2021) | CVPR 2021 | **85.4** | 78.4 | 95.2 | 79.2 | 84.5 |
| DSU (Li et al., 2022) | ICLR 2022 | 83.6 | 79.6 | 95.8 | 77.6 | 84.1 |
| VDN | Ours | 84.3 | **79.8** | 94.6 | **82.8** | **85.4** |

## 4.2 PACS

**Settings:** PACS (Li et al., 2017) is a benchmark for domain generalization task collected from four different domains: photo, art painting, sketch, and cartoon with relatively large domain gaps. Following the widely used setting in Carlucci et al. (2019), we only used the official split of the training set to train the model and all the images from the target domain are used for the test. RmsProp is used to train all the networks with an initial learning rate of 5e-5 without momentum and decrease the learning rate by a factor of 10 at the 50th epoch. The batch size is set to 24 and we sample the same number of images from each domain at the training phase. All the images are cropped to $224 \times 224$ for training and the data augmentations including random crop with a scale factor of 1.25, amplitude mix (Xu et al., 2021) , and random horizontal flip. Other augmentations such as random grayscale are not used as it may conceal the true performance of the model by introducing prior knowledge of the target domain. More specifically, the part from the beginning to the second residual block inclusive is regarded as the task-specific encoder $E_c$, and the remaining part acts as the task-specific network $E_t$. The discriminators $D_c$ and $D_x$ are updated once after every 5 updates of other parts in the framework.

**Results:** The results based on Resnet-18 are reported in Table 2. As we can observe, our method outperforms other state-of-the-art methods. Moreover, we observe that we can achieve much better performance in the sketch domain in a large margin compared with other baseline methods. We conjecture the reason that the Sketch domain may contain less domain-specific information. As shown in Fig. 4, shuffling domain-specific features has less impact on image generation in the domain of Sketch. Due to limited space, the results based on Resnet-50 are reported in the Appendix.

## 4.3 mini-DomainNet

**Settings:** We then consider a larger benchmark mini-DomainNet (Zhou et al., 2020c), which is a subset of DomainNet (Peng et al., 2019a) without noisy labels, for evaluation purpose. In mini-DomainNet, there are more than 140k images in total belonging to 4 different domains, namely sketch, real, clipart, and painting.

Table 3: Evaluation of DG on mini-DomainNet benchmark. We repeat the experiments three times and report the average accuracy in the unseen target domain in the table.

| Method | Reference | Clipart | Real | Painting | Sketch | Avg. |
|---|---|---|---|---|---|---|
| DeepAll | - | 62.86 | 58.73 | 47.94 | 43.02 | 53.14 |
| MLDG (Li et al., 2018a) | AAAI 2018 | 63.54 | **59.49** | 48.68 | 43.41 | 53.78 |
| JiGen (Carlucci et al., 2019) | CVPR 2019 | 63.84 | 58.80 | 48.40 | 44.26 | 53.83 |
| MASF (Dou et al., 2019) | NIPS 2019 | 63.05 | 59.22 | 48.34 | 43.67 | 53.58 |
| RSC (Huang et al., 2020) | ECCV 2020 | 64.65 | 59.37 | 46.71 | 42.38 | 53.94 |
| DSU (Li et al., 2022) | ICLR 2022 | 63.17 | 56.03 | 47.46 | 47.14 | 53.45 |
| VDN | Ours | **65.08** | 59.05 | **48.89** | **49.21** | **55.56** |

For the task-specific encoder $E_c$ and task-specific network $E_t$, we use SGD with a momentum of 0.9 and an initial learning rate of 0.005 as the optimizer. For other parts of our framework, RmsProp without momentum is used with the initial learning rate of 0.0001. For the learning rate scheduler towards all the optimizers, we use the same cosine annealing rule (Loshchilov & Hutter, 2016) with the minimum learning rate of 0 after 100 epochs. The batch size is 128 with a random sampler that roughly samples the same number of images in each domain. We consider data augmentations including the random clip with a probability of 0.5, and random crop the data to the size $96 \times 96$ using the scale factor of 1.25. For a fair comparison, we use the same backbone network Resnet-18 for all the methods and the division of $E_c$ and $E_t$ for our model is the same as the setting in PACS. The update frequency of $D_c$ and $D_x$ is the same with PACS that discriminators update once after every 5 updates of other parts in the framework.

**Results:** We compare our method with MLDG (Li et al., 2018a), JiGen (Carlucci et al., 2019), MASF (Dou et al., 2019), RSC (Huang et al., 2020), and DSU (Li et al., 2022). The results are shown in Table 3. As we can observe, we can achieve better performance compared with other baselines, especially in the sketch domain in a large margin. There is an interesting finding that similar to the PACS benchmark, the performance improvement in the sketch domain is huge, but in the real domain, the performance of our method has some degradation. This may reveal the potential inductive bias of the model. However, our method still has the best overall performance.

## 4.4 Ablation study and perceptual results

In this section, we first present the results of the ablation study to illustrate the effectiveness of each component in our proposed method. We further provide some perceptual results of image generation to show the significance of feature disentanglement.

### 4.4.1 Ablation study

We conduct the ablation study using the PACS benchmark. We first explore the effectiveness of each term in the evidence upper bound we proposed in Theorem 1. Then, the impacts of optimization strategies for each term are evaluated.

**Effectiveness of each term:** We first evaluate the effectiveness of optimizing each term we proposed in the EUB. The results are shown in Table 4 where '+①' and '+②' represent that we utilize the information gain term ① defined in equation 8 and the posterior probability term ② defined in equation 4 but without data augmentation respectively. '+DAug' means we use the generated samples from our generator $G$ for data augmentation, and '+FAug' represents that we adopt Fourier-based data augmentation strategy (Xu et al., 2021) to do the data augmentation. The results demonstrate that both the information gain term and posterior probability term can improve the generalization capability of the model. In addition, combining these two together can attain larger performance improvements. The results also demonstrate that using the generated samples as augmented data can effectively improve the generalization ability of the model.

**Impacts of different optimization strategies:** As for the evidence upper bound we proposed, there are different optimization strategies for each term. For instance, reparameterization trick (Kingma & Welling, 2013) is a widely used method to optimize the term ①. In addition, directly optimizing the L1 reconstruction loss is a usual way to generate sharp reconstructed images. We investigate the impacts of different optimization strategies and the results are

Table 4: Ablation study regarding the effectiveness of each term in the evidence upper bound we proposed. The first row is the DeepAll baseline and the last is the complete version of our method.

| +① | +② | +DAug | +FAug | Art | Cartoon | Photo | Sketch | Avg. |
|----|----|-------|-------|-----|---------|-------|--------|------|
| - | - | - | - | 77.0 | 75.9 | **95.5** | 70.3 | 79.5 |
| ✓ | - | - | - | 77.6 | 77.8 | 93.5 | 71.8 | 80.2 |
| - | ✓ | - | - | 79.7 | 78.4 | 94.3 | 75.2 | 81.9 |
| ✓ | ✓ | - | - | 81.5 | 78.2 | 93.8 | 77.6 | 82.8 |
| ✓ | ✓ | ✓ | - | 82.6 | 78.5 | 94.0 | 82.7 | 84.5 |
| ✓ | ✓ | ✓ | ✓ | **84.3** | **79.8** | 94.6 | **82.8** | **85.4** |

(a) art painting      (b) cartoon      (c) photo      (d) sketch

Figure 4: The generated samples in the unseen target domains. The rows from up to down denote the input image provide the task-specific features, reconstructed images, transformed images, and the images that provide the style features, respectively. Note that the input images all come from the unknown target domain.

shown in Table 5 where the ✓one is the strategy we adopt. As we can see, directly using the reparameterization trick to align the high-dimensional features can lead to side effects. In addition, optimizing the perceptual loss can lead to better performance compared with replacing the reconstruction loss in equation 11 with L1 loss.

Table 5: Ablation study regarding the effects of different optimization strategies. '✓': the strategy we use in the paper. 'reparam': the reparameterization trick (Kingma & Welling, 2013). 'L1 loss': replacing the reconstruction loss in equation 11 with L1 loss.

| +① | +② | Art | Cartoon | Photo | Sketch | Avg. |
|----|----|-----|---------|-------|--------|------|
| - | - | 77.0 | 75.9 | **95.5** | 70.3 | 79.5 |
| reparam | - | 74.4 | 77.6 | 92.4 | 68.3 | 78.2 |
| ✓ | - | 77.6 | 77.8 | 93.5 | 71.8 | 80.2 |
| - | L1 loss | 78.2 | 78.3 | 93.9 | 73.4 | 81.0 |
| - | ✓ | **79.7** | **78.4** | 94.3 | **75.2** | **81.9** |

### 4.4.2 Perceptual results

To provide an intuitive way to understand the effect of disentangling, we further give some perceptual results. For limited space, more visualization results are placed in the supplementary materials.

**Generated samples based on source-domain images:** To demonstrate the effectiveness of our method, we first visualize the generated samples in a cross-domain setting that the pairs of input samples are from different source domains on account that the quality of the samples can reflect the accuracy of the estimated posterior probability and the degree of disentanglement. Some of the generated samples are shown in Fig. 3. The visualization results demonstrate that our method can disentangle the domain-specific features and task-specific features well and generate realistic novel samples with high quality and different styles. In addition, we observe that the reconstructed samples

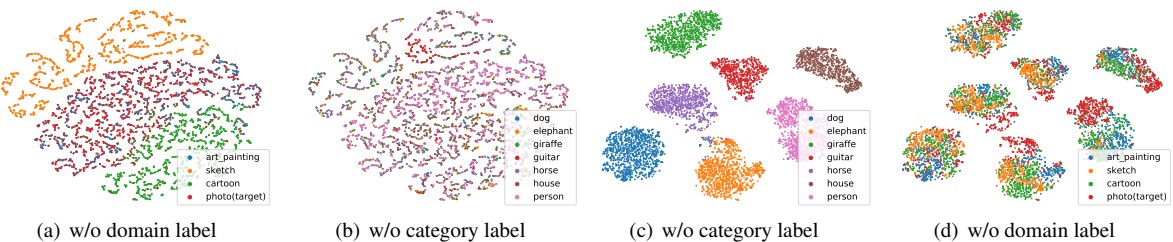

|  |  |  |  |
|---|---|---|---|
| (a) w/o domain label | (b) w/o category label | (c) w/o category label | (d) w/o domain label |

Figure 5: The unsupervised t-SNE visualization results of extracted features from our model. The features in (a,b) are collected from the domain-specific encoder $E_d$ and features in (c,d) are collected from the task-specific encoder $E_c$. PACS benchmark is used for visualization and photo domain is selected as the target domain. (Best viewed in color.)

may not necessarily be the same as the original one, mainly due to the perceptual loss we adopt, as such, we can prevent the latent features from overfitting to the source domain.

**Generated samples based on target-domain images:** To further demonstrate the effectiveness of our proposed framework, we conduct image generation based on the unseen target domain, where the generated samples using the pairs from the same unseen target domain are shown in Fig. 4 based on leave-one-domain-out training manner. From the visualization results, we find that our method can still separate the task-specific features and domain-specific features well even if the networks have never seen the samples from the target domain. More specifically, it can encode the intra-domain style variance based on the observation that the model can generate samples with different styles using the domain-specific features from the same target domain. Meanwhile, the results in Fig. 4 also illustrate that the sketch domain may have little domain-specific features and intra-domain style variance on account that the translated and reconstructed samples are almost the same. This observation further demonstrates the effectiveness of our proposed method by using Sketch as the target domain where a significant performance improvement can be achieved.

### 4.4.3 The visualization of feature embeddings

In addition to the perceptual results of generated samples, we also utilize t-SNE (Van der Maaten & Hinton, 2008) to conduct analysis on the features extracted from two different branches $E_c$ and $E_d$ using our framework by using PACS where Photo is treated as the target domain. The visualization results are shown in Fig. 5 and several findings are observed: in Fig. 5(a) and 5(b), the features from source domains extracted by $E_d$ can be well separated in terms of domain information but cannot be separated in terms of category information. On the contrary, as we can observe from Fig. 5(c) and 5(d), $E_c$ can separate the features in terms of the category information instead of the domain information, which shows the effectiveness of our disentanglement framework.

## 5 Conclusion

In this paper, we propose to tackle the problem of domain generalization from the perspective of variational disentangling. Specifically, we first provide an evidence upper bound regarding the divergence between the distribution of category-specific feature and its invariant ground-truth through Bayes variational inference. Then, we propose an efficient framework to optimize the proposed evidence upper bound for domain generalization. Extensive experiments are conducted to verify the significance of our proposed method. Besides, we conduct experiments of image generation which further justify the effectiveness of our proposed disentanglement framework.

**Limitations:** While we have the same memory usage and speed for inference, our method may increase the training overhead since the existence of auxiliary networks. In addition, due to the instability of GANs, the tuning of the hyperparameters may need some experience and effort. Since our method can be regarded as an unpaired image translation network, it has requirements on the data amount, e.g., too many domains with limited data in each domain may be challenging for our method. Besides, though the information term in our derived evidence upper bound can be exactly calculated, we utilize its substitute given the practical concerns. This could be further improved in the future.

**Broader Impact:** Our proposed method shows the potential in improving the generalization ability of models. It can be used to alleviate the side effect of data bias, e.g., medical images collected from different devices and areas. In addition, our proposed evidence upper bound could give insights to further works. The generated samples with mixing styles and novel characteristics could mitigate stereotypes. However, like other image generation models, our work may also face the risk of being used to generate malicious samples.

## Acknowledgments

This work was carried out at the Rapid-Rich Object Search (ROSE) Lab, Nanyang Technological University, Singapore. This research is supported in part by the Ministry of Education, Republic of Singapore, under its Start-up Grant. This work is also supported by the Research Grant Council (RGC) of Hong Kong through Early Career Scheme (ECS) under the Grant 21200522 and CityU Applied Research Grant (ARG) 9667244.

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

# Contents

# A  Detailed proof

## A.1  Preliminary

Our probabilistic graphical model for classification is represented as Fig. 6. It is notable that the mapping between $z_c$ and $x$, and the mapping between $z_d$ and $x$ are all many-to-many, e.g., there are many plausible images if specify $z_c$ or $z_d$ alone. In addition, given an image $x$, there should exist many possible $z_c$ and $z_d$, i.e., there exist conditional distributions $p(z_c|x)$ and $p(z_d|x)$ instead of a deterministic mapping. Or in other word, given a specific latent code pair $z = [z_c, z_d]$, there exists a conditional distribution $p(x|z)$. This assumption is reasonable since the latent codes $z = [z_c, z_d]$ are usually constrained into a lower-dimensional subspace compared with the image-space so that there may not exist a latent code $z$ that can deterministically correspond to an image $x$.

**Assumption A.1** *(Learnability) Denote the dimension of the image $x$ and the latent feature $z_c$ by $d_x$ and $d_c$, respectively, let $\phi_c : \mathbb{R}^{d_x} \to L^1(\mathbb{R}^{d_c})$ be the mapping between an image $x$ and its probability density function (PDF) $f(z_c|x)$, where $L^1$ denotes 1-norm integrable function space. A deterministic mapping $\phi_g : \mathbb{R}^{d_c} \to \mathbb{N}$ acts as a classifier to predict the category of the image based on its latent feature $z_c$. For a domain generalization problem with $n_s$ source*

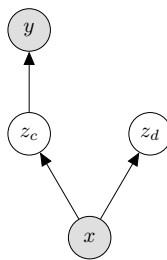

Figure 6: The graphical structure of our proposed model for classification, and the arrows represent learnable mappings. The shaded nodes represent the observed random variables and others are latent random variables. There exist conditional dependences between the image $x$ and the category-specific feature $z_c$ and domain-specific feature $z_d$, respectively. We expect $z_c$ and $z_d$ are independent. In addition, the ground-truth label $y$ only depends on the category-specific feature $z_c$.

*domains ($X_s = X_1 \cup X_2 ... \cup X_{n_s}$) and a target domain $X_t$, we say it learnable if*

$$
\begin{aligned}
&\exists \phi_c, \quad s.t. \\
&\mathbb{E}_{x \sim X_{i|y}}[\phi_c(x)] = \mathbb{E}_{x \sim X_{j|y}}[\phi_c(x)], \forall y \in Y, \forall i, j \in \{1, 2, ..., n_s, t\} \\
&\phi_g(\psi(\phi_c(x))) = y, \forall x \in X_s \cup X_t
\end{aligned}
\tag{13}
$$

*where $X_{i|y}$ represents the conditional distribution of images with the category $y$ in the domain $i$, $\psi(\cdot)$ returns the mean value of the distribution.*

In other words, we say the task is learnable if there exists a domain invariant category-specific encoder $\phi_c$ among source and unseen target domains that can achieve the same marginal distribution $p(z_c|y)$ for each domain. In addition, the extracted category-specific features can be used to correctly predict labels of the images from both source and unseen target domains. This assumption is mild since we only assume that the task is feasible. In addition, we have no additional constraints on the form of the solution.

For a learnable domain generalization task, we aim to find out the domain invariant mapping $\phi_c(\cdot)$ so that we could further train a domain-agnostic classifier. However, given limited data and limited function space, the mapping $\phi_c$ is intractable. To this end, we propose to use an encoder $E_c$ to approximate the unknown ground truth mapping $\phi_c$ by Bayes variational inference. More specifically, we give an upper bound of $\mathcal{D}(Q(\mathbf{z_c}|x)||P(\mathbf{z_c}|x))$ where $P(\mathbf{z_c}|x) = \phi_c(x)$ and $Q(\mathbf{z_c}|x)$ is the conditional distribution embedded by the encoder $E_c$ implemented by a deep neural network. A corresponding framework is further proposed to optimize the proposed evidence upper bound.

Our motivation is summarized as follows

- We first give an evidence upper bound of the KL divergence $\mathcal{D}(Q(\mathbf{z_c}|x)||P(\mathbf{z_c}|x))$ between the approximated conditional distribution of category-specific features $z_c$ and its intractable ground-truth distribution in Lemma A.1 and Theorem A.1.

- We further relax the information gain term of the derived upper bound to make a balance between the two terms in the upper bound. Our relaxed loss is proved to be an upper bound of the original one as shown in Lemma A.2.

- Finally, we demonstrate that under a mild assumption, the upper bound of the divergence $\mathcal{D}(Q(\mathbf{z_c}|x^t)||P(\mathbf{z_c}|x^t))$ in the unseen target domain can also be bounded in Theorem A.2.

## A.2 The upper bound of $\mathcal{D}(Q(\mathbf{z_c}|x)||P(\mathbf{z_c}|x))$

**Lemma A.1** *The KL divergence $\mathcal{D}(Q(\mathbf{z_c}|x)||P(\mathbf{z_c}|x))$ can be represented as*

$$
\mathcal{D}(Q(\mathbf{z_c}|x)||P(\mathbf{z_c})) - E_{z_c \sim Q_{z_c|x}}[\log p(x|z_c)] + \log p(x).
\tag{14}
$$

**Proof A.1**

$$
\begin{aligned}
\mathcal{D}(Q(\mathbf{z_c}|x)||P(\mathbf{z_c}|x)) &= \int q(z_c|x) \log \frac{q(z_c|x)}{p(z_c|x)} \mathrm{d}z_c \\
&= \int q(z_c|x)[\log q(z_c|x) - \log p(z_c|x)] \mathrm{d}z_c \\
&= \int q(z_c|x)[\log q(z_c|x) - \log p(x|z_c) - \log p(z_c) + \log p(x)] \mathrm{d}z_c \\
&= \mathcal{D}(Q(\mathbf{z_c}|x)||P(\mathbf{z_c})) - E_{z_c \sim Q_{z_c|x}}[\log p(x|z_c)] + \log p(x)
\end{aligned}
\tag{15}
$$

*The term $p(x)$ can come out of the expectation $E_{z_c \sim Q_{z_c|x}}$ on account that it does not depend on $z_c$. This completes the proof.*

Based on Lemma 1, we can derive the evidence upper bound of $\mathcal{D}(Q(\mathbf{z_c}|x)||P(\mathbf{z_c}|x))$.

**Theorem A.1** *The evidence upper bound of KL divergence $\mathcal{D}(Q(\mathbf{z_c}|x)||P(\mathbf{z_c}|x))$ between the distribution $Q(\mathbf{z_c}|x)$ and the ground-truth $P(\mathbf{z_c}|x)$ is as follows:*

$$
\underbrace{\mathcal{D}(Q(\mathbf{z_c}|x)||P(z_c))}_{\text{① information gain term}} - \underbrace{E_{z_c \sim Q_{z_c|x}, z_d \sim P_d}[\log p(x|z)]}_{\text{② posterior probability term}} + C
\tag{16}
$$

*where $C$ is a constant, $z = [z_c, z_d]$, and $P_d$ can be an arbitrary prior distribution.*

**Proof A.2**

$$
\begin{aligned}
&\mathcal{D}(Q(\mathbf{z_c}|x)||P(\mathbf{z_c}|x)) \\
&= \underbrace{\mathcal{D}(Q(\mathbf{z_c}|x)||P(\mathbf{z_c})) - E_{z_c \sim Q_{z_c|x}}[\log p(x|z_c)] + \log p(x)}_{\text{Using lemma 1}} \\
&= \mathcal{D}(Q(\mathbf{z_c}|x)||P(\mathbf{z_c})) - E_{z_c \sim Q_{z_c|x}}[\log[\int p(x, z_d|z_c) \mathrm{d}z_d]] + \log p(x) \\
&= \mathcal{D}(Q(\mathbf{z_c}|x)||P(\mathbf{z_c})) - E_{z_c \sim Q_{z_c|x}}[\log[\int \frac{p(x, z_d, z_c)}{p(z_c)}]] + \log p(x) \\
&= \mathcal{D}(Q(\mathbf{z_c}|x)||P(\mathbf{z_c})) - E_{z_c \sim Q_{z_c|x}}[\log[\int \frac{p(z_d)p(z_c|z_d)p(x|z)}{p(z_c)}]] + \log p(x) \\
&= \mathcal{D}(Q(\mathbf{z_c}|x)||P(\mathbf{z_c})) - E_{z_c \sim Q_{z_c|x}}\{\log[E_{z_d \sim P_d} \frac{p(z_c|z_d)}{p(z_c)}p(x|z)]\} + \log p(x) \\
&\leq \mathcal{D}(Q(\mathbf{z_c}|x)||P(\mathbf{z_c})) - \underbrace{E_{z_c \sim Q_{z_c|x}, z_d \sim P_d} \log[\frac{p(z_c|z_d)}{p(z_c)}p(x|z)]}_{\text{Using Jensen's inequality on account that} -\log \text{ is convex}} + \log p(x) \\
&= \mathcal{D}(Q(\mathbf{z_c}|x)||P(\mathbf{z_c})) - \underbrace{E_{z_c \sim Q_{z_c|x}, z_d \sim P_d}[\log p(x|z_c, z_d)]}_{p(z_c|z_d) = p(z_c) \text{ because } z_c \text{ and } z_d \text{ are independent}} + \log p(x) \\
&= \underbrace{\mathcal{D}(Q(\mathbf{z_c}|x)||P(\mathbf{z_c}))}_{\text{① information gain term}} - \underbrace{E_{z_c \sim Q_{z_c|x}, z_d \sim P_d}[\log p(x|z)]}_{\text{② posterior probability term}} + C
\end{aligned}
\tag{17}
$$

*This completes the proof.*

**Assumption A.2** *For an image $x$ there exists a constant $M$ that satisfies $\frac{q(x|z_c)}{q(x)} < M$ given any $z_c$.*

The Assumption A.2 is mild due to the following reasons. First, $x$ is generated based on both $z_c$ and $z_d$. Therefore, the mapping between $z_c$ and $x$ is not deterministic, i.e., $q(x|z_c)$ can not be the impulse function $\delta$. Second, when the random variables $z_c$ and $z_d$ extracted from two encoders $E_c$ and $E_d$ are disentangled in some degree, *i.e.,*

$\sup_{z_c,z_d} \frac{q(z_c|z_d)}{q(z_c)} < k$ where $k$ is a constant, we can obtain the following upper bound

$$
\begin{aligned}
\frac{q(x|z_c)}{q(x)} &= \int \frac{q(x,z_d|z_c)}{q(x)} \mathrm{d}z_d \\
&= \int \frac{q(z_d)q(z_c|z_d)q(x|z_c,z_d)}{q(z_c)q(x)} \mathrm{d}z_d \\
&\leq k E_{z_d \sim Q_{z_d}} \frac{q(x|z_c,z_d)}{q(x)} \\
&\leq k \sup_{z_c,z_d} \frac{q(x|z_c,z_d)}{q(x)} = M
\end{aligned}
\tag{18}
$$

where $q(x|z_c,z_d)$ is the PDF of a predefined Laplace/Gaussian distribution so that it is bounded, and $q(x)$ is a constant for a given $x$. Thus, $\frac{q(x|z_c)}{q(x)}$ is bounded when $z_c$ and $z_d$ is disentangled to some degree. In addition, Eq. equation 18 also demonstrates that the better the disentanglement, the tighter upper bound we can obtain.

**Lemma A.2** *The upper bound of the KL divergence $\mathcal{D}(Q(\mathbf{z_c}|x)||P(\mathbf{z_c}))$ based on Assumption A.2 can be represented as*

$$
M\mathcal{D}(Q(\mathbf{z_c})||P(\mathbf{z_c})) + M\log M. \tag{19}
$$

**Proof A.3**

$$
\begin{aligned}
\mathcal{D}(Q(\mathbf{z_c}|x)||P(\mathbf{z_c})) &= \int q(z_c|x) \log \frac{q(z_c|x)}{p(z_c)} \mathrm{d}z_c \\
&= \int \frac{q(x|z_c)q(z_c)}{q(x)} \log \frac{q(x|z_c)q(z_c)}{q(x)p(z_c)} \mathrm{d}z_c \\
&\leq M[\int q(z_c) \log \frac{q(z_c)}{p(z_c)} \mathrm{d}z_c + \log M] \\
&= M\mathcal{D}(Q(z_c)||P(z_c)) + M\log M
\end{aligned}
\tag{20}
$$

The upper bound given in Lemma A.2 is used to balance the weight of two terms in the upper bound of $\mathcal{D}(Q(\mathbf{z_c}|x)||P(\mathbf{z_c}|x))$ given in Theorem A.1. It is difficult to accurately estimate the posterior probability term in Eq. 16 since the high dimensionality of the image and limited data. On the contrary, the calculation of $\mathcal{D}(Q(\mathbf{z_c}|x)||P(\mathbf{z_c}))$ is trivial and accurate, so it leads to a stable gradient. If we do not relax the constraint of the information gain term in Eq. 16, the gradient accumulation will cause a over-sparse embedding of $z_c$. The ablation study in the main paper also demonstrate the necessity of the relaxation. To further verify that optimizing $\mathcal{D}(Q(\mathbf{z_c})||P(\mathbf{z_c}))$ can help the minimization of $\mathcal{D}(Q(\mathbf{z_c}|x)||P(\mathbf{z_c}))$, we provide a synthetic toy example as follows.

We consider a classical problem, exclusive or, which aims to predict the outputs of XOR logic gates. More specifically, in our setting, we assume $x \in \mathbb{R}^3$ and its label $y = (x_0 > 0) \oplus (x_1 > 0) \oplus (x_2 > 0)$ where $x_i \sim U(-1,1)$ represent the $i$th dimension of $x$. For networks, we use the same backbone network for all experiments which includes three fully connected (FC) layers with the (input dimension, output dimension) as follows $(3,3), (3,2), (2,1)$. For each fully connected layer except the last one, there is a ReLU activation layer following it. We adapt the reparameterization trick (Kingma & Welling, 2013) as shown in Fig. 7 so that we could explicitly obtain the distribution $Q(\mathbf{z_c}|x)$, i.e., the KL divergence between $Q(\mathbf{z_c}|x)$ and the standard Gaussian distribution can be explicitly calculated. For the baseline method, we directly optimize the network by minimizing the cross entropy loss. For the method with the regularization term $L_{reg}$, we additionally utilize a multilayer perceptron (MLP) with three FC layers as the discriminator $D_c$. We keep other hyper-parameters and settings the same. The experimental results are shown in Fig. 8. As we can see, the model can achieve slightly higher test accuracy by introducing $L_{reg}$. In addition, even if we do not directly minimize $\mathcal{D}(Q(\mathbf{z_c}|x)||P(\mathbf{z_c}))$, it can be effectively optimized by minimizing $\mathcal{D}(Q(\mathbf{z_c})||P(\mathbf{z_c}))$ compared with the baseline method. The result in Fig. 8(b) further supports our Lemma A.2 empirically.

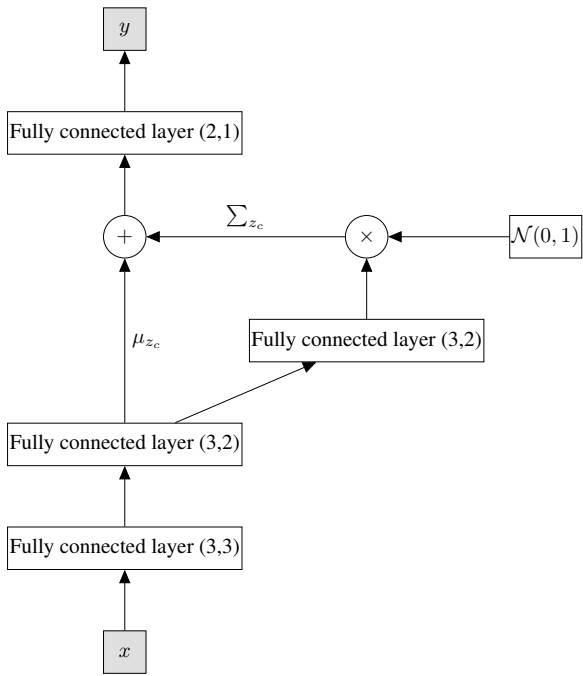

Figure 7: The structure of the network for the toy synthetic experiment. The activation layers are omitted for concise. We utilize the reparameterization trick so that we could explicitly obtain the distribution $Q(\mathbf{z_c}|x)$

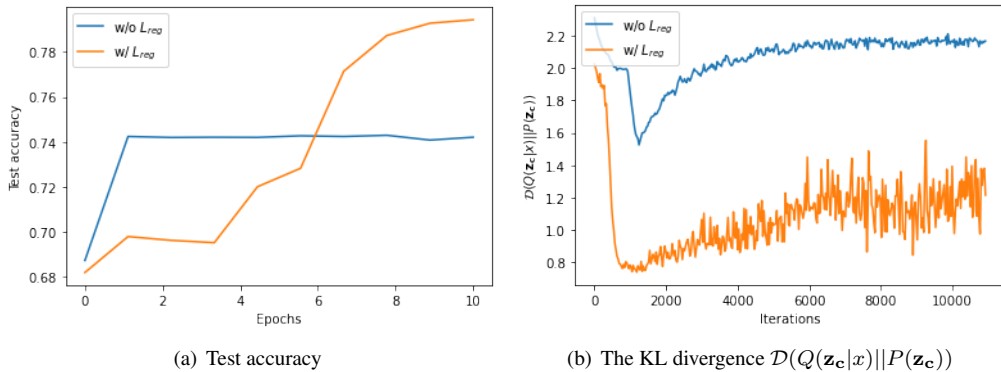

(a) Test accuracy

(b) The KL divergence $\mathcal{D}(Q(\mathbf{z_c}|x)||P(\mathbf{z_c}))$

Figure 8: The comparison of the methods w/ and w/o $L_{reg}$.

### A.3 The upper bound in the unseen target domain

**Assumption A.3** *Given a sample $x^t$ from the target domain $X_t$, there exists a non-empty feasible set $\mathcal{I}$ which is defined as*

$$\mathcal{I} = \{I|q(z_c|x^t) \leq \sum_{i \in I} \beta_i q(z_c|x_i^s), \forall z_c \in \mathbb{R}^{d_c}\} \cap \{I|\phi_c(x^t) = \phi_c(x_i^s), \forall i \in I\},$$

*where $I$ is the index set, $x_i^s$ denotes an arbitrary sample with index $i$ in any source domains, and $q(z_c|x)$ is the probability density function value of $z_c$ conditioned on $x$ from distribution $Q(\mathbf{z_c}|x)$.*

Since we target to deal with the target domain samples with only domain shift, i.e., the samples with unseen $z_d$ and known $z_c$. If the unseen sample has quite a different ground-truth distribution of the feature $z_c$, we cannot guarantee the behavior of the classifier even if we have an ideal classifier $E_c$ that can perfectly disentangle $z_c$ and $z_d$. We assume the task is feasible, i.e., there exist samples in source domains that have the same ground-truth conditional distribution $P_{z_c|x}$. In other words, the second set in Assumption 2 is not empty if the task is feasible.

For the first set in Assumption 2, as long as we assume $q(z_c|x)$ is a distribution that satisfies $q(z_c|x) > 0$ given any $z_c$, e.g., Gaussian/Laplace distribution, for any source domain image set in the second set $\{I|\phi_c(x^t) = \phi_c(x_i^s), \forall i \in I\}$, there always exist a vector of $\beta_i$ that makes $q(z_c|x^t) \leq \sum_{i \in I} \beta_i q(z_c|x_i^s), \forall z_c \in \mathbb{R}^{d_c}$ hold. Therefore, the fesasible set $\mathcal{I}$ is non-empty if the task is feasible.

**Theorem A.2** *Based on Assumption A.3, the KL divergence between $Q(\mathbf{z_c}|x^t)$ and the unknown domain-invariant ground truth distribution $P(\mathbf{z_c}|x^t)$ can be bounded as follows*

$$\mathcal{D}(Q(\mathbf{z_c}|x^t)||P(\mathbf{z_c}|x^t)) \leq \inf_{I \in \mathcal{I}} [\sum_{i \in I} \beta_i \mathcal{D}(Q(\mathbf{z_c}|x_i^s)||P(\mathbf{z_c}|x_i^s))].$$

The above theorem demonstrates that the KL divergence between $Q(\mathbf{z_c}|x)$ and $P(\mathbf{z_c}|x)$ from source domains constitutes the divergence upper bound in the unseen target domain. Therefore, it further supports the rationale and effectiveness of our method. Since no general upper exists in KL divergence, our derived upper bound will also inevitably exist arbitrarily large values. However, it still provides some insights which are intuitive:

- We can get a tighter bound if we have more diverse samples. By increasing the number of source domain samples, we are likely to obtain a larger feasible set to achieve a smaller infimum.

- The better disentangle ability the encoder $E_c$ has, the more similar $q(z_c|x^t)$ and $q(z_c|x_i^s)$ will be. Then $\beta$ are expected to be samller.

## B Detailed architectures of the networks

To be more clear about the architecture of our framework, we give a simplified diagram about the relationship between class-specific encoder $E_c$, domain-specific encoder $E_d$ and generator $G$. More details about the architecture of each network are elaborated in the following sub-sections.

### B.1 The architectures of $E_c$ and $E_t$

#### B.1.1 The architectures of $E_c$ and $E_t$ for Digits-DG

We use ConvNet as the backbone for Digits-DG and divide it into two parts. The details about the division for $E_c$ and $E_t$ are illustrated in Table 6.

#### B.1.2 The architectures of $E_c$ and $E_t$ for PACS and mini-Domainnet

For PACS and mini-Domainnet, we use Resnet-18 as the backbone network and adopt the same division regarding $E_c$ and $E_t$. The details about the division for $E_c$ and $E_t$ are illustrated in Table 7.

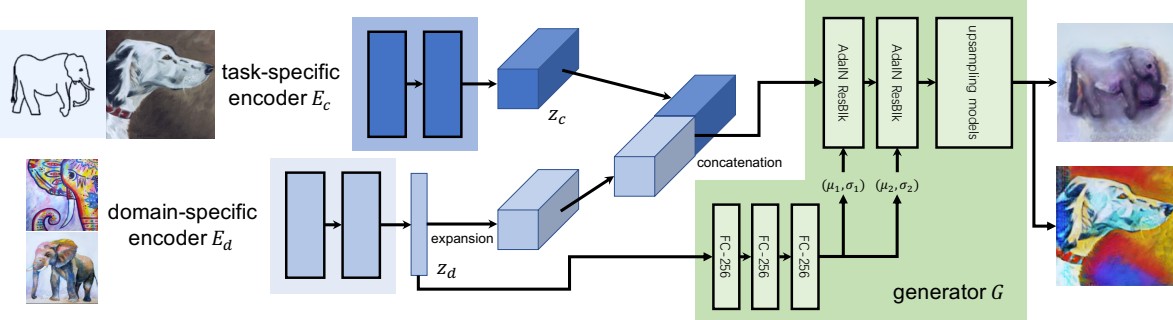

Figure 9: A simplified diagram regarding $E_c$, $E_d$ and $G$. The input of the generator $G$ consists of two parts: the output of the task-specific encoder $z_c$ and the output of the domain-specific encoder $z_d$. The feature $z_d$ is fed into a three-layer network to obtain the mean and variance used by AdaIN ResBlk following Liu et al. (2019). In addition, we reshape $z_d$ to the same size with the feature map of $z_c$ and then concatenate them together.

|  | # | Layer |
|---|---|---|
| task-specific encoder $E_c$ | 1 | Conv2D(in=3, out=64, kernel_size=3, stride=1,padding=1) |
|  | 2 | Relu |
|  | 3 | MaxPool2D(kernel_size=2) |
|  | 4 | Conv2D(in=64, out=64, kernel_size=3, stride=1,padding=1) |
|  | 5 | Relu |
|  | 6 | MaxPool2D(kernel_size=2) |
|  | 7 | Conv2D(in=64, out=64, kernel_size=3, stride=1,padding=1) |
|  | 8 | Relu |
| task-specific network $E_t$ | 9 | MaxPool2D(kernel_size=2) |
|  | 10 | Conv2D(in=64, out=64, kernel_size=3, stride=1,padding=1) |
|  | 11 | Relu |
|  | 12 | MaxPool2D(kernel_size=2) |
|  | 13 | flatten the feature |
|  | 14 | Fully connected layer(in=64*2*2, category number) |

Table 6: The architecture of task-specific encoder $E_c$ and task-specific network $E_t$ for Digits-DG. They are obtained by separating the backbone ConvNet into two parts.

|  | # | Layer |
|---|---|---|
| task-specific encoder $E_c$ | 1 | Conv2D(in=3, out=64, kernel_size=7, stride=2,padding=3) |
|  | 2 | BatchNorm2d |
|  | 3 | Relu |
|  | 4 | MaxPool2d(kernel_size=3,stride=2,padding=1) |
|  | 5 | Relu |
|  | 6 | layer1 $\begin{bmatrix} 3 \times 3 & 64 \\ 3 \times 3 & 64 \end{bmatrix}$ |
|  | 7 | layer2 $\begin{bmatrix} 3 \times 3 & 128 \\ 3 \times 3 & 128 \end{bmatrix}$ |
| task-specific network $E_t$ | 8 | layer3 $\begin{bmatrix} 3 \times 3 & 256 \\ 3 \times 3 & 256 \end{bmatrix}$ |
|  | 9 | layer4 $\begin{bmatrix} 3 \times 3 & 512 \\ 3 \times 3 & 512 \end{bmatrix}$ |
|  | 10 | global average pooling |
|  | 11 | Fully connected layer(in=512, category number) |

Table 7: The architecture of the task-specific encoder $E_c$ and the task-specific network $E_t$ for PACS and mini-DomainNet. They are obtained by separating the backbone Resnet-18 into two parts.

## B.2    The architecture of the domain-specific encoder $E_d$

For all the experiments, we use the same domain-specific encoder. The details of the architecture are shown in Table 8. On account that we use a global average pooling layer to reduce the size of the feature map to $1 \times 1$, the domain-specific code $z_d$ will lose the spatial information and hence only embed the global appearance which is relevant to the domain.

| # | Layer |
|---|---|
| 1 | Conv2D(in=3, out=64, kernel_size=7, stride=1,padding=3) |
| 2 | Relu |
| 3 | Conv2D(in=64, out=128, kernel_size=4, stride=2,padding=1) |
| 4 | Relu |
| 5 | Conv2D(in=128, out=256, kernel_size=4, stride=2,padding=1) |
| 6 | Relu |
| 7 | Conv2D(in=256, out=256, kernel_size=4, stride=2,padding=1) |
| 8 | Relu |
| 9 | global average pooling |
| 10 | Conv2D(in=256, out=128, kernel_size=1, stride=1,padding=0) |

Table 8: The architecture of the domain-specific encoder $E_d$.

## B.3    The architecture of the discriminator $D_x$

PatchGan discriminator (Isola et al., 2017) is adopted as $D_x$ for all experiments. More specifically, the discriminator $D_x$ which is responsible to distinguish real and generated images includes one convolutional layer and 8 activation first residual blocks (Mescheder et al., 2018) as shown in Table 9.

Our discriminator is not only capable of distinguishing real and fake, but it can also distinguish whether the image comes from the specified domain (the number of output channel of our discriminator is the domain number as illustrated in Table 9). If $z_c$ and $z_d$ are not disentangled well, the arbitrarily combined pairs may not generate realistic images with corresponding domain labels, i.e., the disentanglement is supervised by the posterior probability term.

| # | Layer |
|---|-------|
| 1 | Conv2D(in=3, out=64, kernel_size=7, stride=1,padding=3) |
| 2 | ActFirstResBlock(in=64, out=128, activation=leakyRelu, norm=None) |
| 2 | ActFirstResBlock(in=128, out=128, activation=leakyRelu, norm=None) |
| 3 | ReflectionPad2d(padding=1) |
| 4 | AvgPool2d(kernel_size=3, stride=2) |
| 5 | ActFirstResBlock(in=128, out=256, activation=leakyRelu, norm=None) |
| 6 | ActFirstResBlock(in=256, out=256, activation=leakyRelu, norm=None) |
| 7 | ReflectionPad2d(padding=1) |
| 8 | AvgPool2d(kernel_size=3, stride=2) |
| 9 | ActFirstResBlock(in=256, out=512, activation=leakyRelu, norm=None) |
| 10 | ActFirstResBlock(in=512, out=512, activation=leakyRelu, norm=None) |
| 11 | ReflectionPad2d(padding=1) |
| 12 | AvgPool2d(kernel_size=3, stride=2) |
| 13 | ActFirstResBlock(in=512, out=1024, activation=leakyRelu, norm=None) |
| 14 | ActFirstResBlock(in=1024, out=1024, activation=leakyRelu, norm=None) |
| 15 | ReflectionPad2d(padding=1) |
| 16 | AvgPool2d(kernel_size=3, stride=2) |
| 17 | LeakyRelu(1024) |
| 18 | Conv2D(in=1024, out=domain number, kernel_size=1, stride=1,padding=1) |

Table 9: The architecture of the discriminator $D_x$. **ActFirstResBlock**: activation first residual blocks (Mescheder et al., 2018)

## B.4   The architecture of the discriminator $D_c$

The discriminator $D_c$ is built by multiple fully connected layers. The details are illustrated in Table 10. For the fake sample, the input of the discriminator $D_c$ is the task-specific feature $z_c$ after average-pooling and is concated with the domain label which acts as supervised information. For the real one, the input is the random variable sampling from a standard Gaussian distribution and concated with a random domain label.

| # | Layer |
|---|-------|
| 1 | Linear(in=input dim, out=512, bias=True) |
| 2 | Relu |
| 3 | Dropout(probability=0.2) |
| 4 | Linear(in=512, out=512, bias=True) |
| 5 | Relu |
| 6 | Dropout(probability=0.2) |
| 7 | Linear(512, 2) |
| 8 | activation layer $g_f(v) = -\exp(-v)$ |

Table 10: The architecture of the discriminator $D_c$. **input dim**: the number of source domains + the channel number of the feature $z_c = E_c(x)$

## B.5   The architecture of the generator $G$

We use roughly the same architecture of the generator $G$ for Digits-DG and PACS/mini-DomainNet. The difference is mainly in the number of channels and the number of upsampling modules. The architecture of the generator is shown in Table 11. Note that the three-layer network for AdaIN ResBlk (Huang et al., 2018) is not included in the table.

(a) The generator for Digits-DG

| # | layer |
|---|---|
| 1 | AdaIN ResBlk(192) |
| 2 | AdaIN ResBlk(192) |
| 3 | Upsample(scale_factor=2, nearest) |
| 4 | Conv2D(in=192, out=96, kernal_size=5, s=1, p=2) |
| 5 | InstanceNorm |
| 6 | Relu |
| 7 | Upsample(scale_factor=2, nearest) |
| 8 | Conv2D(in=96, out=48, kernal_size=5, s=1, p=2) |
| 9 | InstanceNorm |
| 10 | Relu |
| 11 | Conv2D(in=48, out=3, kernal_size=5, s=1, p=2) |
| 12 | Tanh & Normalize |

(b) The generator for PACS and mini-DomainNet

| # | layer |
|---|---|
| 1 | AdaIN ResBlk(256) |
| 2 | AdaIN ResBlk(256) |
| 3 | Upsample(scale_factor=2, nearest) |
| 4 | Conv2D(in=256, out=128, kernal_size=5, s=1, p=2) |
| 5 | InstanceNorm |
| 6 | Relu |
| 7 | Upsample(scale_factor=2, nearest) |
| 8 | Conv2D(in=128, out=64, kernal_size=5, s=1, p=2) |
| 9 | InstanceNorm |
| 10 | Relu |
| 11 | Upsample(scale_factor=2, nearest) |
| 12 | Conv2D(in=64, out=32, kernal_size=5, s=1, p=2) |
| 13 | InstanceNorm |
| 14 | Relu |
| 15 | Conv2D(in=32, out=3, kernal_size=5, s=1, p=2) |
| 16 | Tanh & Normalize |

Table 11: The architecture of the generator $G$ except the three-layer network for AdaIn ResBlk (Liu et al., 2019).

# C  Extra results

## C.1  Results on PACS benchmark using Resnet-50 backbone

| Algorithm | A | C | P | S | Avg |
|---|---|---|---|---|---|
| ERM | $83.2 \pm 1.3$ | $76.8 \pm 1.7$ | $\mathbf{97.2} \pm 0.3$ | $74.8 \pm 1.3$ | 83.0 |
| IRM | $81.7 \pm 2.4$ | $77.0 \pm 1.3$ | $96.3 \pm 0.2$ | $71.1 \pm 2.2$ | 81.5 |
| GroupDRO | $84.4 \pm 0.7$ | $77.3 \pm 0.8$ | $96.8 \pm 0.8$ | $75.6 \pm 1.4$ | 83.5 |
| Mixup | $85.2 \pm 1.9$ | $77.0 \pm 1.7$ | $96.8 \pm 0.8$ | $73.9 \pm 1.6$ | 83.2 |
| MLDG | $81.4 \pm 3.6$ | $77.9 \pm 2.3$ | $96.2 \pm 0.3$ | $76.1 \pm 2.1$ | 82.9 |
| CORAL | $80.5 \pm 2.8$ | $74.5 \pm 0.4$ | $96.8 \pm 0.3$ | $78.6 \pm 1.4$ | 82.6 |
| MMD | $84.9 \pm 1.7$ | $75.1 \pm 2.0$ | $96.1 \pm 0.9$ | $76.5 \pm 1.5$ | 83.2 |
| DANN | $84.3 \pm 2.8$ | $72.4 \pm 2.8$ | $96.5 \pm 0.8$ | $70.8 \pm 1.3$ | 81.0 |
| CDANN | $78.3 \pm 2.8$ | $73.8 \pm 1.6$ | $96.4 \pm 0.5$ | $66.8 \pm 5.5$ | 78.8 |
| MTL | $85.6 \pm 1.5$ | $78.9 \pm 0.6$ | $97.1 \pm 0.3$ | $73.1 \pm 2.7$ | 83.7 |
| SagNet | $81.1 \pm 1.9$ | $75.4 \pm 1.3$ | $95.7 \pm 0.9$ | $77.2 \pm 0.6$ | 82.3 |
| ARM | $\mathbf{85.9} \pm 0.3$ | $73.3 \pm 1.9$ | $95.6 \pm 0.4$ | $72.1 \pm 2.4$ | 81.7 |
| VREx | $81.6 \pm 4.0$ | $74.1 \pm 0.3$ | $96.9 \pm 0.4$ | $72.8 \pm 2.1$ | 81.3 |
| RSC | $83.7 \pm 1.7$ | $82.9 \pm 1.1$ | $95.6 \pm 0.7$ | $68.1 \pm 1.5$ | 82.6 |
| VDN(Ours) | $85.8 \pm 0.6$ | $\mathbf{83.5} \pm 0.7$ | $96.7 \pm 0.3$ | $\mathbf{85.6} \pm 0.6$ | 87.9 |

Table 12: Evaluation of domain generalization performance on PACS benchmark using ResNet-50 backbone. The mean value and corresponding standard error of the test accuracy are reported.

We further conduct experiments using ResNet-50 backbone and report the mean accuracy and std in the target domain when finish the training. The results of 5 repeated experiments are shown in Table 12 and the results demonstrate that our method achieves better performance than SOTA methods[4].

## C.2  Parameter sensitivity

We explore the influence of two important hyper-parameters $\lambda_{rec}$ and $\lambda_{gan}$. We evaluate the performance of models trained with different $\lambda_{rec}$ and $\lambda_{gan}$ while keeping other settings the same on the sketch domain of PACS. The results are shown in Fig 10. As we can see, the performance remains relatively stable among different hyper-parameters. We empirically find that $\lambda_{gan}$ has a relatively larger impact on the performance compared with $\lambda_{rec}$. We conjecture the reason is that an inappropriate $\lambda_{gan}$ may cause unnatural artifacts. For a larger $\lambda_{rec}$, it imposes a stronger regularization on $z_c$ and $z_d$ so it also causes performance degradation.

## C.3  Results on Camelyon17

We further evaluate the effectiveness of our method on the WILDS (Koh et al., 2021; Sagawa et al., 2021) benchmark. More specifically, we utilize Camelyon17 (Bandi et al., 2018) dataset which is collected from different hospitals. Each hospital is regarded as a domain and the domain variations are mainly in the data collection and processing. We use the official train/val/test split of the dataset and report the test accuracy when we achieved the highest accuracy on the evaluation set. We use DenseNet121 as the backbone which is the same for all competitors. The generated samples are shown in Fig. 11 and the accuracy in the unseen target domain is reported in Table 13. As we can see in Fig. 11, our method can well disentangle the domain variations, e.g., the color difference caused by the processing while preserving the necessary details. In addition, the quantitative result exhibits huge improvement compared with the baseline methods, which further demonstrates the effectiveness of our method.

## C.4  Computational cost

For inference, we do not need the auxiliary networks so that we have the same computational cost and the clock time as the baseline method.

---

[4]The results of other methods are from the camera ready version of the paper "In search of lost domain generalization", ICLR 2021.

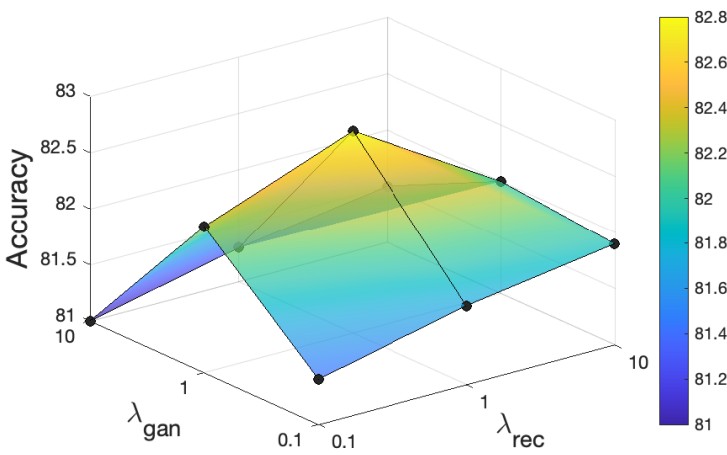

Figure 10: Sensitivity analysis of hyper-parameters $\lambda_{rec}$ and $\lambda_{gan}$.

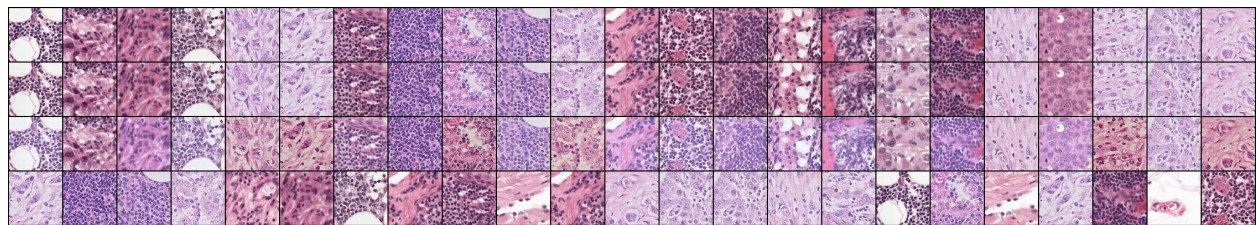

Figure 11: The generated samples in source domains using Camelyon17 (Bandi et al., 2018) dataset. The images in the first row are the input images that we want to keep the category information and in the last row are the input images that provide the style information. The second row is the reconstructed one and the third row is the translated one.

|  | Evaluation Accuracy | Test Accuracy |
|---|---|---|
| ERM | 85.8 (1.9) | 70.8 (7.2) |
| IRM | 86.2 (1.4) | 64.2 (8.1) |
| VDN(Ours) | 88.0 (2.4) | 90.3 (3.1) |

Table 13: Evaluation of domain generalization performance on Camelyon17 dataset using DenseNet-121 backbone. The mean value and corresponding standard error of the test accuracy are reported.

For training, we compare our method with the baseline method and MLDG using the settings of the experiments on Camelyon17 as illustrated in Sec. C.3. All the experiments are conducted on the same server with RTX A5000s. We use the same settings, e.g., the same batch size and image size, for all experiments. The results are reported in Table 14. As we can see in the table, our method has an acceptable computational cost, which is similar to the previous work MLDG (Li et al., 2018a). It is worth noting that, the extra memory usage of our method is mainly due to the existence of data augmentation, i.e., the batch size of our method can be regarded as twice that of other methods.

|  | Parameters | Memory usage | Speed (s/iter) |
|---|---|---|---|
| Baseline | 6,955,906 | 2739MB | 0.1153 |
| MLDG (Li et al., 2018a) | 6,955,906 | 4013MB | 0.4604 |
| VDN (Ours) | 8,408,844 | 5765MB | 0.2115 |

Table 14: The computational cost of our method for training. We train models using mixed precision for all experiments using the same platform.

## C.5 Extra visualization results

Besides the image generation task in the source domains or in the target domain that we have shown the results in the paper, we are also interested in the "cross-domain" performance, i.e., from source domains to unseen target domain and from unseen target domain to source domains. Assuming that we have two images $x_i$ and $x_j$ and their corresponding task-specific features $z_{c_i}$, $z_{c_j}$ and domain-specific features $z_{d_i}$, $z_{d_j}$, the translated images with different mixture degrees are generated using $G(z_{c_i}, \lambda z_{d_i} + (1-\lambda)z_{d_j})$ where $\lambda$ in $[0.0, 0.1, 0.2, ..., 1.0]$. Note that in this work, we do not use the generated samples with a mixture style for data augmentation nor use the discriminator to supervise the generated images with a mixture style. How to effectively utilize the sample of mixed style may be a potential research direction.

### C.5.1 From the target domain to source domains

We first investigate the performance of our framework that translates the samples from the unseen target domain to source domains. The results are shown in Fig. 14. As we can observe, even if the framework has never seen the samples in the unseen target domain, it can still extract task-specific features that are compatible with the domain-specific features from source domains and then generate the samples with the style of source domains. This further verifies the robustness and generalization ability of our method.

### C.5.2 From source domains to the target domain

We then explore the performance of our proposed framework that translates the image from source domains to the unseen target domain. The visualization results are shown in Fig. 15. As we can see, the framework may be difficult to generate samples of the unseen target domain. This is reasonable because our framework has never seen the sample distribution of the target domain and it is difficult to generate OOD samples with large domain gaps. However, our proposed framework can generate samples from a new domain, i.e., the image that is not exactly the same as the sketch domain but has some similar characters such as having little task-irrelevant information.

### C.6 Comparison with other data augmentation-based DG methods

It is still an open problem and a research hotspot to generate realistic images. The conditional generation task or unpaired image translation task usually requires large-scale data or data which are not very diverse. For example, CycleGan uses 939 horse images and 1177 zebra images to train a model that can only transfer the horse image to zebra even if they have relatively high similarity. However, the PACS dataset we used only has 1670 images in the *photo* domain and there are both different domains and different categories in it. In addition, *photo* domain images have more details, especially compared with *sketch* so it will be more difficult to learn and generate.

Most importantly, we aim to conduct the classification task instead of the generation task. The generated images are only side products of our method and the perceptual quality of generated images is beyond the scope of our paper.

In addition, our ablation study demonstrates that by using our generated samples as augmented training data, the generalization ability of the model can be further improved and outperform other data generation-based DG methods.

Here we display the visualization results reported in their paper from the methods that also aim to generate novel data using the same PACS dataset in Fig. 12. As we can see, these methods fail to generate data with large divergence, e.g., the translation to *sketch* domain in Borlino et al. (2021).

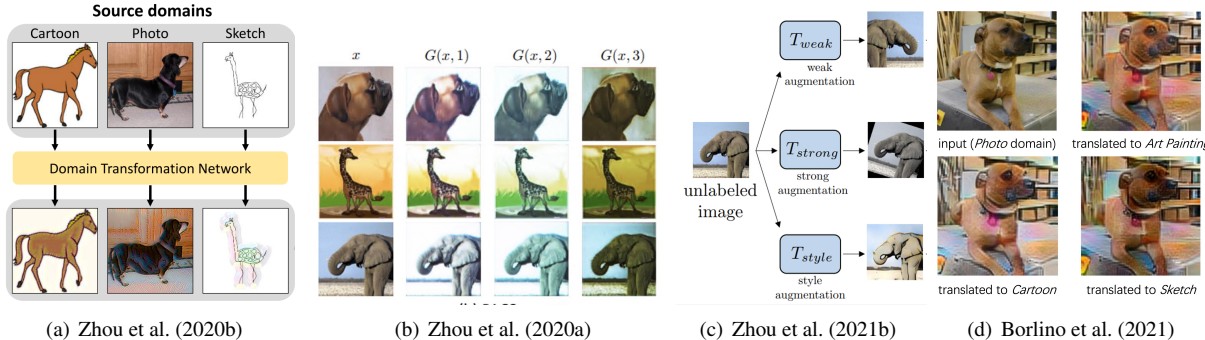

(a) Zhou et al. (2020b)  (b) Zhou et al. (2020a)  (c) Zhou et al. (2021b)  (d) Borlino et al. (2021)

Figure 12: Comparison with other DG methods based on novel data generation. Best zoom in for details.

## C.7 Failure cases

Some failure cases of our proposed model are displayed in Fig. 13. For the Digits-DG benchmark, we may fail to generate samples with the style that multiple digits exist in a single image. We conjecture that it may be caused by the inductive bias of our network structure, e.g., the utilization of the AdaIn layer. For the natural image, our model may fail to transfer images from the sketch domain to the real photo/art painting domains, which include much more information. This may be due to the limitation of the data amount. While some of the samples do not have desirable perceptual quality, they may still include some important features of the desired category, so they are still valuable for acting as augmented data.

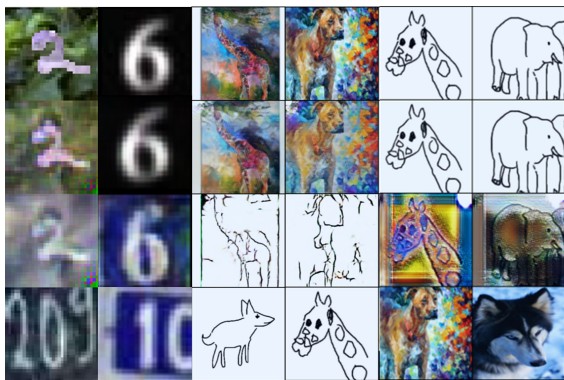

Figure 13: Some failure cases of our model.

# D Experiment details

## D.1 Experiment environment

Parts of the experiments are conducted on a Windows workstation with Ryzen 5900X, 64GB RAM, and an Nvidia RTX 3090. Others are conducted on a Linux server with Intel(R) Xeon(R) Silver 4210R CPU, 256GB RAM, and RTX 2080-TIs. For the software, PyTorch 1.9 is used. We do not find a significant difference of trained models under two different environments.

## D.2   Tuning strategy for hyper-parameters

Since we have multiple hyper-parameters, we elaborate on the tuning strategy here for reference.

- We first explore the split strategy of $E_c$ and $E_t$. If $E_c$ is too small, then there will be little semantic information. While if $E_c$ is very deep, then we find it is difficult to reconstruct the image using a feature map with small width and height given a small but diverse dataset.

- After finding the appropriate division of $E_c$ and $E_t$ that can well reconstruct the image, we then explore the hyper-parameter of the regularization term $\lambda_{reg}$ since it is a relatively individual hyper-parameter. After finding the $\lambda_{reg}$ that can increase the generalization ability, we fix it for the following experiments. We also find $\lambda_{rec}$ that does not degrade the classification performance much and can well reconstruct the images, and then fix it for the following experiments.

- We then introduce the discriminator into the training. We find that the updating frequency of the generator and discriminator is significant. We first find the updating frequency that will not cause model collapse and then finetune $\lambda_{gan}$.

- We search the hyper-parameters on the PACS benchmark since it is a relatively small dataset. We find that the hyper-parameters used in PACS exhibit good robustness and can also be used in other benchmarks.

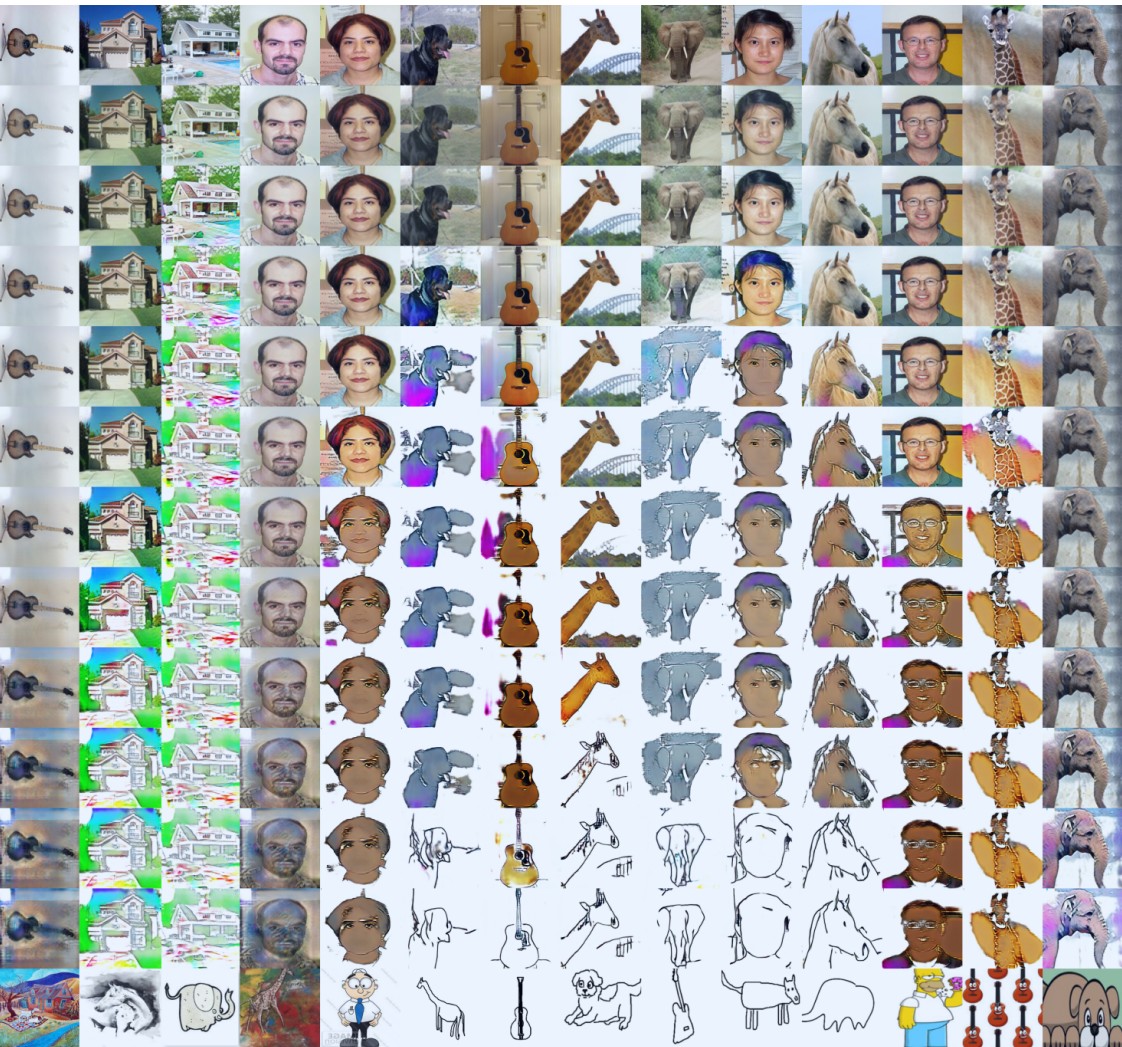

Figure 14: The visualization results of the image translation from the unseen target domain to source domains. The photo domain in PACS is used as the unseen target domain. The first row is the input image from the unseen target domain that provides the task-specific features and the last row is from source domains that provide the domain-specific features. Other rows are images that are generated images $G(z_{c_i}, \lambda z_{d_i} + (1 - \lambda)z_{d_j})$ using different mixture degrees.

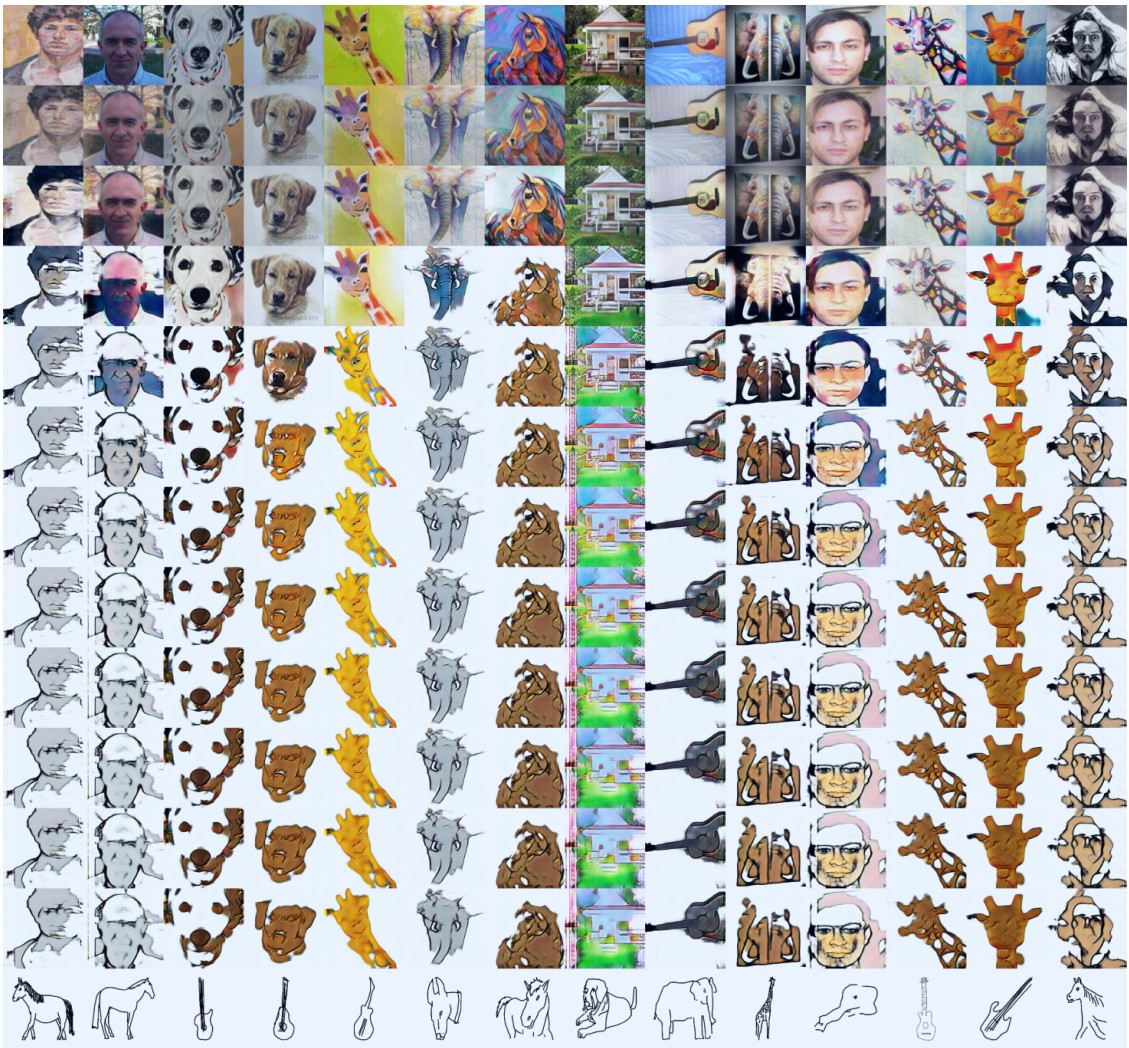

Figure 15: The visualization results of the image translation from source domains to the uneen target domain. The domain sketch in PACS is used as the unseen target domain. The first row is the input image from source domains that provide the task-specific features and the last row is from the unseen target domain that provides the domain-specific features. Other rows are images $G(z_{c_i}, \lambda z_{d_i} + (1 - \lambda)z_{d_j})$ using different mixture degree.

