# OpenReview forum: "Variational Disentanglement for Domain Generalization"
_TMLR — Accepted by TMLR_

### Review · Reviewer_DDX9 · 2022-06-02

**Summary Of Contributions:**

This paper addressed the OOD (or domain generalization) problem through the ​​disentanglement and variational method.  Specifically, they aim to disentangle style and semantic information, then the framework could both generate new images and improve the prediction accuracy on the unseen domain. Extensive empirical results validated the framework.


**Broader Impact Concerns:**

N/A. Authors are encouraged to make a short statement to illustrate the societal benefits (e.g., mixing style could mitigate stereotype) or risks (e.g, inducing bias) of the proposed framework.

**Requested Changes:**

Overall this paper has multiple merits, however, several parts lack discussions/analysis/proper organizations. Based on the aforementioned reviews and guidelines of TMLR, I would suggest the following main revisions:

1. Paper organization/writing. Authors are suggested to revise them. (Q1-12)
2. Theory and practice gap. Authors are suggested to include discussions or adjust their claims within the paper, to accurately illustrate the contributions. (Q1-6)
3. Practical aspects. Ideally, *authors are not required to conduct additional experiments but adding relevant discussions/analysis*. As for the failure images, authors are encouraged to propose analysis from the current failure results (if there are).(Q2-4, Q1 for the discussion propose)


**Strengths And Weaknesses:**

## Update after discussion

Authors have properly addressed all major concerns. Thus I would recommend acceptance.

## Summary:

### Strong points:
a. [This paper is interesting to the community.] Briefly, it proposed a practical and principled probabilistic framework for addressing domain generalization. Moreover, the framework could generate new style-transfer images. In contrast, the conventional approaches generally did the data augmentation on the embedding space and could not generate real-images.

b. [Theoretical aspects] I would appreciate the authors for clearly illustrating the theoretical assumptions (e.g, assumption 1,2) within the paper. It is quite beneficial to know the limitations and scope.

c. [Practical aspects] The extensive empirical results demonstrate the effectiveness of the proposed approach. Specifically I like the results in Appendix C2.2, which clearly depicts empirical benefits of the proposed approach.

### Weak points:

a. [Paper organization/writing] In general, the paper is feasible to follow, while some parts are confusing/unclear or augments are inaccurate.

b. [Theory and practice gap] It is appreciated that authors could provide several theoretical analyses, while there exists several non-ignorable gaps between theory and practice. Following the guidelines of TMLR, authors are required to claim/adjust such gaps.

c. [Practical concerns] The high-level idea sounds correct to me, while there are several practical concerns.

## Detailed comments on weak points
### Paper organization/writing.

1. [Sec 1]  “Generally, existing DG research works can be categorized into two streams, invariant feature representation learning and data augmentation.” In general the meta-learning based approach could be a third stream.
2. [Sec 1] “they may face the risk of overfitting to the source domains without carefully specifying the invariant information to learn.” I would refer to a recent paper [1] for discussing the overmatching risk to the source domains, which provides a formal justification about this concern.
3. [Sec 1]”To further optimize the upper bound, we develop an efficient framework named Variational Disentanglement Network (VDN)”. I think it should be effective rather than efficient, since the efficiency has not been justified in the paper.
4. [Sec 1] “we propose to optimize the category-specific feature zc and domain-specific feature zd at the same time.” I would suggest category(label)-specific feature zc.
5. [Sec 1] “Meanwhile, by further minimizing our proposed information term which is to filter out redundant information, zc is expected to have less domain-specific information”. I think it is safe to remove this since zc is defined as independent to the domain.
6. [Sec 1] “Our proposed evidence upper bound further supports the rationality and builds the tie between previous feature alignment-based methods (Li et al., 2018b) and disentangle-based methods.” It seems I missed something, but I could not find evidence how they are related. This contribution might be adjusted/better illustrated.
7. [Sec 3] “all source domains are represented as Xi , Yi and X, Y respectively”. Seems like a broken sentence. What is X,Y ? the target ? Besides, the conventional notation assumes X,Y as random variables and P(X), P(Y) as distributions.
8. [Assumption 1] $\Phi$ is the means of the random variable, I would suggest avoiding using this function since it is not clear in the definition. It could be simply expressed as the expectation notation $E_{x} \phi_c(x)$.
 9. [Lemma 1, Theory 1, Theory 2] The definition of x is consistent with Assumption 1? I.e $x \in X_s \cup X_t$ ?
10. [Sec 3.2] “following the core idea of F-GAN” -> f-GAN, f is a convex function here.
11. [Paper Organization] The proposed loss, the algorithm as well as the network structure are suggested to be better formed. This reviewer could follow the whole protocol while it requires checking back and forth between different components. Authors are suggested to clearly present the whole losses in Sec 3.1.1, and provide explanations in the following sections.
12. [Eq (9)] It should be $D_x$ rather than D. Am I correct?

### Theory and practice gap

1. [Assumption 1] Learnability. I agree this assumption is mild but it is suggested to clearly discuss limitations. As far as I understand, it has two limitations (1) It requires the embedding distribution is noise free such that there exists a deterministic labeling function to recover label y from the embedding.(e.g, see the theoretical result in [2]) (2) it implicitly assumes the label distribution should be uniform. (e.g, see the theoretical result in [3]).
2. [Assumption 2] I have difficulty in understanding Assumption 2. (1) The $I$ set should be the source samples that have the sample covariate w.r.t. the target in the embedding space. I.e need to find all $z_s^{i} = z_t$ to formulate such a set. This assumption in the embedding space sounds reasonable for me since it is a low dimensional data but it is still a bit difficult since $z_s$ is a continuous random variable. (2) The variational distribution w.r.t. $Z_c$ should be upper bound ( $\beta$ is positive????) by the corresponding source samples. Upon reflection, I understand that there always exists such a coefficient $\beta$ to guarantee an upper bound. But it is preferable that the author would state the intuitions behind assumption 2.
3. [Theory 2]  The derivation seems straightforward for me. However, the bound could be vacuous if the $\beta$ is arbitrarily large in assumption 2. (e.g, set $\beta$ = 10^6), which induces a non-meaningful OOD generalization bound. Thus a discussion is expected herein.
4. [Eq (3)] The posterior loss seems not a strong tie with the task, rec and gans loss. Clearly, the posterior loss is purely unsupervised, the variational term does not include the label information. The authors are suggested to make clear illustration between the posterior loss and rec and gans loss. (hint: The prediction loss could be introduced by the information bottleneck/information projection principle to maximize the correlation between $z_c$ and $y$. Or express a joint variable $(x,y)$ in the variational inference. )
5. [Appendix A.1] The graph seems inaccurate. I think it should be in the anti-causal direction. I.e, y -> z_c -> x <- z_d, then the generative model could model this problem without considering the shift of P(y). This is also aligned with Assumption 1.
6. [Assumption A.2] I could not understand how q(x|z_c) is bounded. As far as I understand,
$$q(x|z_c) / q(x) = q(x|z_c)/\sum_{z_c} q(z)q(x|z_c) \leq 1/q(z_c)=M,$$
thus it should be q(z_c) is positive within its distribution support. Since the upper bound depends on M, thus the upper bound could be involved with the prior distribution q(z_c) and computationally feasible (consider q(z_c) being gaussian for example).

### Practical aspects
1. If I understand correctly, I think the proposed loss could be fully unsupervised. Is it possible to generate the interpolated samples in Appendix C.2.2 without supervision? This question is for the discussion propose, no additional experiments are required.
2. It seems the proposed approach has multiple hyper-parameters. The authors are suggested to illustrate the tuning strategy.
3. [Tab.12] The PACS (S) is surprisingly better than all other baselines (>10%), are there any possible explanations?
4. As for generated images (C2.2 in appendix), is it possible to illustrate the failure results? I.e some images are poorly generated. I think it would be quite interesting to see such results.

Ref

[1] On the benefits of representation regularization in invariance based domain generalization. Machine Learning Journal, 2022.

[2] Domain adaptation with conditional distribution matching and generalized label shift. NeurIPS 2020

[2] Aggregating From Multiple Target-Shifted Sources. ICML 2021

---

> ### Author Response · Authors · 2022-07-03
> **response (part 5)**
>
> [10] **“following the core idea of F-GAN” -> f-GAN, f is a convex function here.**:
>
> We change F-GAN to f-GAN.
>
> [11] **[Paper Organization] The proposed loss, the algorithm as well as the network structure are suggested to be better formed. This reviewer could follow the whole protocol while it requires checking back and forth between different components. Authors are suggested to clearly present the whole losses in Sec 3.1.1, and provide explanations in the following sections.**
>
> Thanks for the advice. In the revised version, we first give the definition of each loss term in Sec. 3.3.1 (Overall framework), so that it will be more convenient to read.
>
> [12] **[Eq (9)] It should be D_x rather than D. Am I correct?**
>
> Thanks for finding the typo. It is indeed $D_x$.

---

> ### Author Response · Authors · 2022-07-03
> **response (part 4)**
>
> [6] **[Sec 1] “Our proposed evidence upper bound further supports the rationality and builds the tie between previous feature alignment-based methods (Li et al., 2018b) and disentangle-based methods.” It seems I missed something, but I could not find evidence how they are related. This contribution might be adjusted/better illustrated.**
>
> In "Domain generalization with adversarial feature learning (Li et al., 2018b)", the author proposes to "introduce a prior distribution to regularize the distribution of the mapped source domains data in the feature
> space using an adversarial training procedure." They assume the prior distribution $p(h)$ is Laplace distribution, and propose to optimize the divergence $JSD(p(h)||p(Q(x)))$ by adversarial training, i.e., optimizing $J_{gan}$ in their paper in Eq. (2). Our regularization term is quite similar to it while we provide an interpretation from the perspective of Bayes inference.
>
> We add a footnote in our paper to explain the relationship as follows
> > In Li et al., 2018b, the author empirically proposes to use an adversarial autoencoder to minimize the divergence between $p_h$ and $p_x$ where $p_h$ is the latent code distribution and $p_x$ is the prior distribution(Laplace distribution in the paper).
>
> [7，9] **[Sec 3] “all source domains are represented as Xi , Yi and X, Y respectively”. Seems like a broken sentence. What is X,Y ? the target ? Besides, the conventional notation assumes X,Y as random variables and P(X), P(Y) as distributions.**
>
> **[Lemma 1, Theory 1, Theory 2] The definition of x is consistent with Assumption 1?**
>
>  We unify the definition of the symbols in the section preliminary, e.g., $X_i$ represents the image set from the $i_{th}$ domain so they are compatible with the following contents.
>
> [8] **$\psi$ is the means of the random variable, I would suggest avoiding using this function since it is not clear in the definition. It could be simply expressed as the expectation notation.**

---

> ### Author Response · Authors · 2022-07-03
> **response (part 3)**
>
>
> [4] **[Eq (3)] The posterior loss seems not a strong tie with the task, rec and gans loss. Clearly, the posterior loss is purely unsupervised, the variational term does not include the label information. The authors are suggested to make clear illustration between the posterior loss and rec and gans loss. (hint: The prediction loss could be introduced by the information bottleneck/information projection principle to maximize the correlation between and. Or express a joint variable in the variational inference. )**
>
> Our model is not purely unsupervised as illustrated at the beginning (part 1 of the response).
>
> [5] **[Appendix A.1] The graph seems inaccurate. I think it should be in the anti-causal direction. I.e, y -> z_c -> x <- z_d, then the generative model could model this problem without considering the shift of P(y). This is also aligned with Assumption 1.**
>
> The generative model can be indeed modeled as $y->z_c->x<-z_d$. While in Appendix A.1, we aim to describe the causal model of our image classification process like Fig. 2(a) in [a].
>
> [a] Domain generalization using causal matching, ICML 2021
>
> We emphasize the graphical structure is for classification in the revised paper in Appendix A.1, for example
> > The graphical structure of our proposed model *for classification and the arrows represent learnable mappings.*
>
> in the title of Fig. 6.
>
> [6] **[Assumption A.2] I could not understand how q(x|z_c) is bounded. As far as I understand, $q(x|z_c) / q(x) = q(x|z_c)/\sum_{z_c} q(z)q(x|z_c) \leq 1/q(z_c)=M,$ thus it should be q(z_c) is positive within its distribution support. Since the upper bound depends on M, thus the upper bound could be involved with the prior distribution q(z_c) and computationally feasible (consider q(z_c) being gaussian for example).**
>
> Since given a specific image $x$, $q(x)$ is a constant, i.e., $q(x|z_c)$ is bouned is equivalent to the assumption that $\frac{q(x|z_c)}{q(x)}$ is bounded.  In Eq. (17), we further show that it is equivalent to a mild assumption that $\sup_{z_c, z_d}\frac{q(z_c|z_d)}{q(z_c)}$ is bounded. For example, for the extreme case that $z_c$ and $z_d$ are well disentangled, its upper bound is $1$.
>
> For clarity, in our revised version, we rewrite the assumption as follows
> > For an image $x$ there exists a constant $M$ that satisfies $\frac{q(x|z_c)}{q(x)} <M$ given any $z_c$.
>
>
>
> ## Practical aspects
>
> [1] **If I understand correctly, I think the proposed loss could be fully unsupervised. Is it possible to generate the interpolated samples in Appendix C.2.2 without supervision? This question is for the discussion propose, no additional experiments are required.**
>
> As illustrated in the _Posterior probability term_ at the beginning, our model is not unsupervised.
>
> [2] **It seems the proposed approach has multiple hyper-parameters. The authors are suggested to illustrate the tuning strategy.**
>
> We add the following content regarding the tuning strategy for hyper-parameters to Appendix D.2 in the revisied version.
>
> > For implementation, the strategies are as follows
> > - We first explore the split strategy of $E_c$ and $E_t$. If $E_c$ is too small, then there will be little semantic information. While if $E_c$ is very deep, then we find it is difficult to reconstruct the image using a feature map with small width and height given a small but diverse dataset.
> > - After finding the appropriate division of $E_c$ and $E_t$ that can well reconstruct the image, we then explore the hyper-parameter of the regularization term $\lambda_{reg}$ since it is a relatively individual hyper-parameter. After finding the $\lambda_{reg}$ that can increase the generalization ability, we fix it for the following experiments. We also find $\lambda_{rec}$ that does not degrade the classification performance much and can well reconstruct the images, and then fix it for the following experiments.
> >
> > - We then introduce the discriminator into the training. We find that the updating frequency of the generator and discriminator is significant. We first find the updating frequency that will not cause model collapse and then finetune $\lambda_{gan}$.
> >
> > - We search the hyper-parameters on the PACS benchmark since it is a relatively small dataset. We find that the hyper-parameters used in PACS exhibit good robustness and they can also be used in other benchmarks.

---

> ### Author Response · Authors · 2022-07-03
> **response (part 2)**
>
> ## Theory and practice gap
>
> [1] **[Assumption 1] Learnability. I agree this assumption is mild but it is suggested to clearly discuss limitations. As far as I understand, it has two limitations (1) It requires the embedding distribution to be noise-free such that there exists a deterministic labeling function to recover label y from the embedding. (e.g, see the theoretical result in [2]) (2) it implicitly assumes the label distribution should be uniform. (e.g, see the theoretical result in [3]).**
>
> In assumption 1, we indeed assume the embedding distribution is noise-free. We change the statement in Sec 3.1 from
>
> > - Such an assumption is mild and is commonly adopted in the community of domain generalization (Muandet et al., 2013).
> >
> > to
> >
> > - While we assume a noiseless case similar to (Tachet des Combes et al., 2020), such an assumption is mild and is
> commonly adopted in the community of domain generalization (Muandet et al., 2013).
>
>
> In addition, We could extend Assumption 1 to the noise existence setting naturally. For example, in our current setting, we aim to approximate the unknown clean conditional distribution. Similarly, we can approximate the noise embedding that can achieve the minimum loss.
>
> Similar to other works of DG, we make an implicit assumption that the label distributions between source and target domains are not varied a lot. We add the following statement to Assumption 1 in the paper.
>
> > Similar to other DG works (Li et al., 2018d;c; Hu et al., 2020), there is an implicit assumption that the label prior does not vary a lot among
> source and target domains. The cases that label priors are largely different may in the scope of imbalanced classification (Sun et al., 2009) and heterogeneous domain generalization (Wang et al., 2020c; Li et al., 2019b).
>
> [2] **[Assumption 2] I have difficulty in understanding Assumption 2. (1) The set should be the source samples that have the sample covariate w.r.t. the target in the embedding space. I.e need to find all to formulate such a set. This assumption in the embedding space sounds reasonable for me since it is low dimensional data but it is still a bit difficult since is a continuous random variable. (2) The variational distribution w.r.t. should be upper bound (is positive????) by the corresponding source samples. Upon reflection, I understand that there always exists such a coefficient to guarantee an upper bound. But it is preferable that the author would state the intuitions behind assumption 2.**
>
> We add the following statement in the main paper after assumption 2.
> > Assumption 2 is a mild assumption since it holds as long as the task is feasible (the second set is not empty) and $q(z_c|x)$ is a distribution that satisfies $q(z_c|x)>0$ given any $z_c$, e.g., Gaussian distribution (the first set is not empty).
>
> We add the following more detailed explanation to Appendix A.3 to explain why the feasible set $\mathcal{I}$ is non-empty.
>
> > Since we target to deal with the target domain samples with only domain shift, i.e., the samples with unseen $z_d$ and known $z_c$. If the unseen sample has quite a different ground-truth distribution of the feature $z_c$, we cannot guarantee the behavior of the classifier even if we have an ideal classifier $E_c$ that can perfectly disentangle $z_c$ and $z_d$. We assume the task is feasible, i.e., there exist samples in source domains that have the same ground-truth conditional distribution $P_{z_c|x}$. In other words, the second set in Assumption 2 is not empty if the task is feasible.
> >
> > For the first set in Assumption 2, as long as we assume $q(z_c|x)$ is a distribution that satisfies $q(z_c|x)>0$ given any $z_c$, e.g., Gaussian/Laplace distribution, for any source domain image set in the second set $\{I | \phi_c(x^t) = \phi_c(x_i^s) , \forall i\in I\}$, there always exist a vector of $\beta_i$ that makes $q(z_c|x^t) \leq \sum_{i\in I} \beta_i q(z_c|x_i^s), \forall z_c \in \mathbb{R}^{d_c}$ hold. Therefore, the fesasible set $\mathcal{I}$ is non-empty if the task is feasible.
>
>
>
> [3] **[Theory 2] The derivation seems straightforward for me. However, the bound could be vacuous if the  is arbitrarily large in assumption 2. (e.g, set  = 10^6), which induces a non-meaningful OOD generalization bound. Thus a discussion is expected herein.**
>
> We add the following discussion to Appendix A.3.
>
> > Since no general upper exists in KL divergence, our derived upper bound will also inevitably exist arbitrarily large values. However, it still provides some insights which are intuitive:
> >- We can get a tighter bound if we have more diverse samples. By increasing the number of source domain samples, we are likely to obtain a larger feasible set to achieve a smaller infimum.
> >- The better disentangle ability the encoder $E_c$ has, the more similar $q(z_c|x^t)$ and $q(z_c|x^s_i)$ will be. Then $\beta$ are expected to be smaller.

---

> ### Author Response · Authors · 2022-07-03
> **response (part 1)**
>
> Thanks for your prompt and detailed comments. The revised contents are colored in blue in the new version of the paper.
>
> **Posterior probability term**
>
> Our posterior term is not unsupervised. Since, we aim to maximize $p(x|z_c, z_d)$ instead of $p(x)$, a realistic image with an incorrect domain and category label should not be assigned a high likelihood by the discriminator $D_x$.
>
> For the implementation of domain-label guidance, the number of output channels in the last layer of our discriminator $D_x$ is the domain number as illustrated in Appendix B.3. More specifically, assume for an image $x$ with domain index $i$, we use the $i$-th channel of the discriminator output as the final output, so it can be regarded as explicitly using the domain label as supervision.
>
> For $y$, we can naturally follow the design of the conditional GAN to model the conditional distribution. For the digits-dg benchmark, $z_c$ is concatenated as a part of the discriminator input as illustrated in Appendix B.4. Therefore, the conditional distribution is directly estimated. For other benchmarks, we find using a classification loss on $z_c$ is good enough. It does not increase the training load and simplifies the architecture of the discriminator. Therefore, for PACS and mini-DomainNet, we assume if $z_c$ corresponds to the correct category label, then the generated samples will also have the desired category. We empirically find this is effective and reasonable since $z_c$ carries more information than $z_d$ due to the network design.
>
> We are sorry that the definition in Eq.(8) was unclear. We adopt the following modification
> > $
> p(x|z_c,z_d) = \\left\\{\\begin{matrix}
> La(x|G(z_c,z_d), \beta)) & \\text{w/ ground-truth} \\\
> D_x(G(z_c,z_d);z_{c_i}, z_{d_j}) & \\text{w/o ground-truth}
> \\end{matrix}\\right.
> $
> >
> > where $La$ denotes the PDF of Laplace distribution (corresponding to the L1 norm) to measure the pixel reconstruction loss, and $D_x(\cdot| z_c, z_d)$ is the estimation from the discriminator that can be based on the corresponding category and domain labels/features. To optimize the term Eq. (8),  $D_x$ needs to have the capability to distinguish between the real and generated images $G(z_{c_i}, z_{d_j})$ through random combined latent features, and to differentiate whether the samples have the desired domain and category. Meanwhile, the generator $G$ is expected to produce realistic outputs.

---

> > ### Comment · Reviewer_DDX9 · 2022-07-05
> > **Post-rebuttal**
> >
> > Dear authors,
> >
> > Thanks for your detailed responses and revisions! Most of my concerns have been clarified and it remains only two *minor* points:
> >
> > - I understand that loss is supervised. While in the theoretical derivation, the theory does not explicitly consider the label information. (e.g, Lemma 1, Theory 1, the Z_c is implicitly determined by label y). This is a minor point, while it is expected to clearly illustrate in the paper such as in the appendix.
> > - About label distribution shift in Assumption 1. I agree with your point and I would suggest a bit more discussion. If the label distribution P(y) is largely different across the sources, the upper bound is related to the discrepancy of label distribution (Theorem 2 in paper [1]) and conditional distribution shift in the embedding space (assumption 1).
> >
> > Ref: [1] Aggregating From Multiple Target-Shifted Sources. ICML 2021

---

> > > ### Author Response · Authors · 2022-07-07
> > > **Post-rebuttal**
> > >
> > >
> > > Thanks for your effort to help us improving the quality of the paper. The changes are colored in green in the newly uploaded version.
> > >
> > > **[1] I understand that loss is supervised. While in the theoretical derivation, the theory does not explicitly consider the label information. (e.g, Lemma 1, Theory 1, the Z_c is implicitly determined by label y). This is a minor point, while it is expected to clearly illustrate in the paper such as in the appendix.**
> > >
> > > We add the following discussion after the definiation of Theremon 1 in the papaer
> > >
> > > - It is worth noting that $z_c$ and $z_d$ are implicitly determined by the category and domain label respectively, i.e., only the realistic image in accord with its corresponding category and domain could have a high probability density value.
> > >
> > > **[2] About label distribution shift in Assumption 1. I agree with your point and I would suggest a bit more discussion. If the label distribution P(y) is largely different across the sources, the upper bound is related to the discrepancy of label distribution (Theorem 2 in paper [1]) and conditional distribution shift in the embedding space (assumption 1).**
> > >
> > > We add more discussion as follows:
> > >
> > > - In addition, theoretical analysis can be found in Shui et al (2021) which demonstrates that the upper bound of target risk is related to the discrepancy of label distribution
> > >
> > > Ref: Aggregating From Multiple Target-Shifted Sources. ICML 2021

---

### Review · Reviewer_AP9E · 2022-06-08

**Summary Of Contributions:**

The work develops a framework for domain generalization. The idea is to rely on domain-specific information and category-specific information through feature disentanglement. Specifically, given a multitude of source domains, the framework learns a domain invariant embedding for classification and domain-specific features in a variational setting. The schematic framework can be seen in Fig. 2 and it involves the task-specific encoder, the domain-specific encoder, a decoder (‘generator’) and two discriminators. A number of experiments in small datasets are conducted.

**Broader Impact Concerns:**

There is no broader impact statement in the paper at the moment. And indeed, similarly to image translation, this framework could have negative impacts, so those need to be described in the revised version.

**Requested Changes:**

The quality of writing can be substantially improved, as there are multiple mistakes both in language and consistency errors. For instance, no title has a “:” symbol apart from sec. 4.4.3. There is space required in the phrase “the target domain.Meanwhile” (page 11). There are several errors that proofreading would fix, e.g. “all the datasets may has”, “target domain is pass through”, “can also be benefited”. Another error made consistently in the text is the citet package usage. After citet the verb should be in plural form if there are multiple authors, e.g. Li et al. propose. This should be fixed throughout the manuscript. The current writing is not appropriate for a high quality paper.

Is assumption 2 a strict assumption? Could the authors elaborate on the first part of the assumption on why it holds?

What are the limitations of the method? How about the broader impact?

In eq. 9 the objective is max_D min_{E, G}, while for GANs it is usually min_G max_D. Why is it reversed here and are any guarantees lost with this change?
I think one very commonly used dataset is the WILDS benchmark, however the paper does not include it. Is there some reasoning behind excluding it?

In sec. 2 the paper mentions that the proposed framework conducts translation in a very efficient manner. However, in the pix2pix paper cited in the same paragraph some of the mappings are learned with few images; are less images used in the experimental validation of this method?

What is the computational cost of the proposed method with respect to the baselines? Both in terms of parameters and with respect to the clock time.

How sensitive is the proposed method to the various hyper-parameters? An ablation study would help to clarify the important hyper-parameters for the interested practitioner.


**Strengths And Weaknesses:**

strengths:
+ The problem of domain generalization is significant and timely.
+ The work tries to relate it well with the literature.

weaknesses:
- The writing could be improved (see below).
- There is no section on limitations and broader impact at the moment.

---

> ### Author Response · Authors · 2022-07-03
> **response (part 2)**
>
> **[5] I think one very commonly used dataset is the WILDS benchmark, however the paper does not include it. Is there some reasoning behind excluding it?**
>
> We further evaluate the effectiveness of our proposed method on Camelyon17, a dataset of the WILDS benchmark. The results are placed in the Appendix. As we can see in the appendix, our method can well disentangle the category-specific features and domain-specific features. We can obtain samples with the desired target styles while preserving the deterministic features. In addition, we achieve significant accuracy improvement in the unseen target which demonstrates the effectiveness of our method as shown in the following table.
>
> > |  | Evaulation accuracy | Test accuracy |
> > | --- | --- | --- |
> > | ERM | 85.8 (1.9) | 70.8 (7.2) |
> > | IRM | 86.2 (1.4) | 64.2 (8.1) |
> > | VDN(Ours)| **88.0 (2.4)**| **90.3 (3.1)**|
> > Table 13: Evaluation of domain generalization performance on Camelyon17 dataset using DenseNet-121 backbone. The mean value and corresponding standard error of the test accuracy are reported.
>
>
> **[6] In sec. 2 the paper mentions that the proposed framework conducts translation in a very efficient manner. However, in the pix2pix paper cited in the same paragraph some of the mappings are learned with few images; are fewer images used in the experimental validation of this method?**
>
> Our proposed method is for a very DIFFERENT problem setting, compared to Pix2Pix:  Our proposed method is unsupervised, whereas Pix2Pix is a SUPERVISED method, i.e., requires training of PAIRED data, which is much more restrictive in practice. Correspondingly, unsupervised methods usually require a larger-scale training set, due to the lack of pairing and supervision. In addition, the model of Pix2Pix only transfers the image from one domain to another. However, our model needs to have the capacity to transfer from one arbitrary domain to any other domain, which is a many-to-many mapping.
>
> Our method can indeed disentangle and generate relatively high-quality images using a relatively small-scale dataset. For example, in the PACS benchmark, there are less than 9000 images in the training set while there are 4 different domains and 7 categories. We also add a comparison with other data augmentation-based DG methods in the Appendix. Our method can generate samples with better diversity and quality using the same dataset.
>
> **[7] What is the computational cost of the proposed method with respect to the baselines? Both in terms of parameters and with respect to the clock time**
>
> For inference, we do not need the auxiliary networks so that we have the same computational cost and the clock time as the baseline method. We add the following content to the Appendix regarding the training overhead
>
> > For training, we compare our method with the baseline method and MLDG using the settings of the experiments on Camelyon17 as illustrated in Sec. C.2. All the experiments are conducted on the same server with RTX A5000s. We use the same settings, e.g., the same batch size and image size, for all experiments. The results are reported in Table 14. As we can see in the table, our method has an acceptable computational cost, which is similar to the previous work MLDG~\citep{li2018learning}. It is worth noting that, the extra memory usage of our method is mainly due to the existence of data augmentation, i.e., the batch size of our method can be regarded as twice that of other methods.
> > | | Parameters | Memory usage | Speed (s/iter) |
> > | --- | --- | --- | --- |
> > |Baseline | 6,955,906 | 2739MB | 0.1153 |
> > MLDG (Li et al., 2018a) | 6,955,906 | 4013MB | 0.4604 |
> > VDN (Ours) | 8,408,844 | 5765MB | 0.2115 |
> >
> > Table 14: The computational cost of our method for training. We train models using mixed precision for all experiments using the same platform.
>
> **[8] How sensitive is the proposed method to the various hyper-parameters? An ablation study would help to clarify the important hyper-parameters for the interested practitioner.**
>
> Thanks for your suggestions. We add more ablation studies in the appendix. To sum up, we explore the sensitivities of two important hyper-parameters $\lambda_{rec}$ and $\lambda_{gan}$. The results are shown in Fig 10 in the Appendix. We evaluate the performance of models on the sketch domain of PACS. The results in the Appendix demonstrate that the performance remains relatively stable among different hyper-parameters. We empirically find that $\lambda_{gan}$ has a relatively larger impact on the performance compared with $\lambda_{rec}$. We conjecture the reason is that an inappropriate $\lambda_{gan}$ may cause unnatural artifacts. For a larger $\lambda_{rec}$, it imposes a stronger regularization on $z_c$ and $z_d$ so it also causes performance degradation.  In addition, we elaborate on our tuning strategy and add more content regarding how we choose the hyper-parameters in Appendix D.2.

---

> ### Author Response · Authors · 2022-07-03
> **response (part 1)**
>
> Many thanks for the detailed comments. The revised contents are colored in blue in the new version of the paper.
>
> **[1] The quality of writing can be substantially improved, as there are multiple mistakes both in language and consistency errors. For instance, no title has a “:” symbol apart from sec. 4.4.3. There is space required in the phrase “the target domain.Meanwhile” (page 11). There are several errors that proofreading would fix, e.g. “all the datasets may has”, “target domain is pass through”, “can also be benefited”. Another error made consistently in the text is the citet package usage. After citet the verb should be in plural form if there are multiple authors, e.g. Li et al. propose. This should be fixed throughout the manuscript. The current writing is not appropriate for a high quality paper.**
>
> Thanks for the suggestions regarding improving the paper's quality. We fixed the typos that the reviewer pointed out. Besides, the revised manuscript has been improved with better organization and presentation.
>
> **[2] Is assumption 2 a strict assumption? Could the authors elaborate on the first part of the assumption on why it holds?**
>
> We add the following statement in the main paper after assumption 2.
> > Assumption 2 is a mild assumption since it holds as long as the task is feasible (the second set is not empty) and $q(z_c|x)$ is a distribution that satisfies $q(z_c|x)>0$ given any $z_c$, e.g., Gaussian distribution (the first set is not empty).
>
> More details are added to Appendix A.3 to explain why the feasible set $\mathcal{I}$ is non-empty.
>
> > Since we target to deal with the target domain samples with only domain shift, i.e., the samples with unseen $z_d$ and known $z_c$. If the unseen sample has quite a different ground-truth distribution of the feature $z_c$, we cannot guarantee the behavior of the classifier even if we have an ideal classifier $E_c$ that can perfectly disentangle $z_c$ and $z_d$. We assume the task is feasible, i.e., there exist samples in source domains that have the same ground-truth conditional distribution $P_{z_c|x}$. In other words, the second set in Assumption 2 is not empty if the task is feasible.
> >
> > For the first set in Assumption 2, as long as we assume $q(z_c|x)$ is a distribution that satisfies $q(z_c|x)>0,\quad \forall z_c$, e.g., Gaussian/Laplace distribution, for any source domain image set in the second set $\{I | \phi_c(x^t) = \phi_c(x_i^s) , \forall i\in I\}$, there always exist a vector of $\beta_i$ that makes $q(z_c|x^t) \leq \sum_{i\in I} \beta_i q(z_c|x_i^s), \forall z_c \in \mathbb{R}^{d_c}$ hold. Therefore, the fesasible set $\mathcal{I}$ is non-empty if the task is feasible.
>
> Assumption 2 is a mild assumption since the first set is always non-empty under some common and mild assumptions, e.g., q(z_c|x^t) is a distribution that is non-zero over the entire space. A target domain that is less likely to source domains may lead to a looser upper bound. We also add the following discussion to Appendix A.3.
>
> > Since no general upper exists in KL divergence, our derived upper bound will also inevitably exist arbitrarily large values. However, it still provides some insights which are intuitive:
> >- We can get a tighter bound if we have more diverse samples. By increasing the number of source domain samples, we are likely to obtain a larger feasible set to achieve a smaller infimum.
> >- The better disentangle ability the encoder $E_c$ has, the more similar $q(z_c|x^t)$ and $q(z_c|x^s_i)$ will be. Then $\beta$ are expected to be smaller.
>
> **[3] What are the limitations of the method? How about the broader impact?**
>
> We add the following limitations and the broader impact into the conclusion section.
>
> > **Limitations:** While we have the same memory usage and speed for inference, our method may increase the training overhead since the existence of auxiliary networks. In addition, due to the instability of GANs, the tuning of the hyper-parameters may need some experience and effort.
> >
> > **Broader Impact:** Our proposed method shows the potential in improving the generalization ability of models. It can be used to alleviate the side effect of data bias, e.g., medical images collected from different devices and areas. In addition, our proposed evidence upper bound could give insights to further works. The generated samples with mixing styles and novel characteristics could mitigate stereotypes. However, like other image generation models, our work may also face the risk of being used to generate malicious samples.
>
> **[4] In eq. 9 the objective is max_D min_{E, G}, while for GANs it is usually min_G max_D. Why is it reversed here and are any guarantees lost with this change? I**
>
> Thanks for pointing this out. In the revised version, we rewrite the objective of Eq. 9 to the commonly used form $\min_{E_c, E_d, G}\max_{D_x} L_{gan}$.

---

### Review · Reviewer_J5eu · 2022-06-18

**Summary Of Contributions:**

This paper considers disentangling the label information (that is invariant) and domain style for addressing the domain generalization problem better. This idea (in general) could lead more papers to focus on the solvability of the DG problem, which is good for the area. They also first give a theoretical bound and then design their algorithm, making the whole algorithm sound and theoretical solid. The experiments verify that the proposed method works better than existing methods (sota experiments or ablation study). In general, the main idea of this paper and the theory proposed in this paper are interesting. Thus, I would like to give a major revision to this paper.

**Broader Impact Concerns:**

There is no concern about the ethical implications of the work. DG is a well-known and significant problem in the field.

**Requested Changes:**

Cons:

1. Assumption 1 seems incompleted. Please note that conditional distribution is the same does not mean the joint distribution is the same. For example, if the class prior in the target domain is very different from those in the source domains, DG seems also impossible. How do you avoid this? Some remarks should be added regarding this issue.

2. In Assumption 2, why do you use \ge rather than =. q is just a pdf function. It is not easy to understand Assumption 2.

3. Section 3.3.2 and Section 3.3.3 are not easy to follow. More details are required for completeness.

4. In Section 3.3.2, this paper uses D(Q(z_c)||P(z_c)) to replace with D(Q(z_c|x)||P(z_c)). More discussions are need here. Some synthetic experiments may be helpful.

**Strengths And Weaknesses:**

Pros:

1. The whole idea is interesting. DG problem will be impossible if no assumption between domains is made. This paper considers a fundamental assumption regarding DG, i.e., some invariant features exist. Thus, we can consider detaching the label information and style information.

2. The proposed method is based on a developed theoretical bound, which is sound and solid.

3. Experiments verify that the proposed method is effective and better than many sota methods, which further verifies the contribution of the proposed theory and algorithms.

Cons:

1. Assumption 1 seems incompleted. Please note that conditional distribution is the same does not mean the joint distribution is the same. For example, if the class prior in the target domain is very different from those in the source domains, DG seems also impossible. How do you avoid this? Some remarks should be added regarding this issue.

2. In Assumption 2, why do you use \ge rather than =. q is just a pdf function. It is not easy to understand Assumption 2.

3. Section 3.3.2 and Section 3.3.3 are not easy to follow. More details are required for completeness.

4. In Section 3.3.2, this paper uses D(Q(z_c)||P(z_c)) to replace with D(Q(z_c|x)||P(z_c)). More discussions are need here. Some synthetic experiments may be helpful.

---

> ### Author Response · Authors · 2022-07-03
> **response (part 1)**
>
> Thank you for your detailed comments. The revised contents are colored in blue in the new version of the paper.
>
> **1. Assumption 1 seems incompleted. Please note that conditional distribution is the same does not mean the joint distribution is the same. For example, if the class prior in the target domain is very different from those in the source domains, DG seems also impossible. How do you avoid this? Some remarks should be added regarding this issue.**
>
> We agree that the conditional distribution is the same does not mean the joint distribution is the same. Similar to other works of DG, we make an implicit assumption that the label distributions between source and target domains are not varied a lot. We add the following statement to Assumption 1 in the paper.
>
> > Similar to other DG works (Li et al., 2018d;c; Hu et al., 2020), there is an implicit assumption that the label prior does not vary a lot among
> source and target domains. The cases that label priors are largely different may in the scope of imbalanced classification (Sun et al., 2009) and heterogeneous domain generalization (Wang et al., 2020c; Li et al., 2019b).
>
> Different from the vanilla method that directly learns a mapping between $x$ and $y$, we further introduce a latent variable $z_c$ and regularization are conducted on it. The regularization terms conducted on the latent code $z_c$ may alleviate the side-effect of the label bias. For example, in an extreme binary classification, the ratio between positive and negative is 99.9:0.01 in source domains while 1:1 in the target domain. The vanilla method high-likely learns a trivial mapping that always predicts "positive". While for our method, the regularization terms impel the latent code $z_c$ to include necessary information for image reconstruction to avoid it from being meaningless.
>
>
> **2. In Assumption 2, why do you use \ge rather than =. q is just a pdf function. It is not easy to understand Assumption 2.**
>
> Assumption 2 aims to help obtain the upper bound in Theorem 2.
> The reason why we use $\leq$ instead of $=$ in Assumption 2 is that the equation may not hold or it may lead to a looser upper bound. $\leq$ allows us to obtain a larger feasible set to obtain a tighter upper bound.
> We add more discussion in the appendix regarding Assumption 2 as follows
>
> > Since we target to deal with the target domain samples with only domain shift, i.e., the samples with unseen $z_d$ and known $z_c$. If the unseen sample has quite a different ground-truth distribution of the feature $z_c$, we cannot guarantee the behavior of the classifier even if we have an ideal classifier $E_c$ that can perfectly disentangle $z_c$ and $z_d$. We assume the task is feasible, i.e., there exist samples in source domains that have the same ground-truth conditional distribution $P_{z_c|x}$. In other words, the second set in Assumption 2 is not empty if the task is feasible.
> >
> > For the first set in Assumption 2, as long as we assume $q(z_c|x)$ is a distribution that satisfies $q(z_c|x)>0,\quad \forall z_c$, e.g., Gaussian/Laplace distribution, for any source domain image set in the second set $\{I | \phi_c(x^t) = \phi_c(x_i^s) , \forall i\in I\}$, there always exist a vector of $\beta_i$ that makes $q(z_c|x^t) \leq \sum_{i\in I} \beta_i q(z_c|x_i^s), \forall z_c \in \mathbb{R}^{d_c}$ hold. Therefore, the fesasible set $\mathcal{I}$ is non-empty if the task is feasible.
>
> **3. Section 3.3.2 and Section 3.3.3 are not easy to follow. More details are required for completeness.**
>
> We add more details to the methodology part. The modifications are mainly as follows
> > - We give the definitions of each term in the section ``overall framework`` so that readers can have a preliminary understanding of our method.
> > - We give a more accurate definition to the posterior term to make a clear elaboration that our discriminator is based on both domain-label and category label/feature, instead of based on an unsupervised manner.
> > - We change the subsection tiles from "Minimizing the information gain term" to "Estimation of the information gain term", and "Minimizing the posterior probability term" to "Estimation of the posterior probability term". The change is to avoid potentially misleading, i.e., the minimization of these two terms are joint as shown in Algorithm 1 instead of optimizing them separately.

---

> ### Author Response · Authors · 2022-07-03
> **response (part 2)**
>
> **4. In Section 3.3.2, this paper uses D(Q(z_c)||P(z_c)) to replace with D(Q(z_c|x)||P(z_c)). More discussions are need here. Some synthetic experiments may be helpful.**
>
> In summary, the main motivation is that directly optimizng $D(Q(z_c|x)||P(z_c))$ using reparameterization loss imposes a too restrictive regularizer on the latent code $z_c$ given limited data. Instead, we find $D(Q(z_c)||P(z_c))$ can be a good substitution for it. The reasons are mainly twofolds: first, $D(Q(z_c)||P(z_c))$ constitute the upper bound of $D(Q(z_c|x)||P(z_c))$. Second, minizminig  $D(Q(z_c)||P(z_c))$ is found useful empirically in the previos literature.
>
> We add experiments on a synthetic toy dataset to further verify the effectiveness of optimizing the upper bound of $D(Q(z_c|x)||P(z_c))$. In summary, we utilize the reparameterization trick to train the model so that we can explictly calculate the exact value of $D(Q(z_c|x)||P(z_c))$. We compare the curve of $D(Q(z_c|x)||P(z_c))$ w/ and w/o the optimization of $D(Q(z_c)||P(z_c))$. The experimental results show that minimizng $D(Q(z_c)||P(z_c))$ can effectively suppress the value of $D(Q(z_c|x)||P(z_c))$. More details are added to the revised version at the end of Sec A.2 in the Appendix.
>
> More discussions about the replacement can be found in Lemma A.2 in the Appendix.

---

> ### Comment · Reviewer_J5eu · 2022-07-06
> **Thanks for the responses**
>
> I think the authors have addressed all of my concerns. I would like to support this paper for acceptance.

---

> > ### Author Response · Authors · 2022-07-07
> > **Post-rebuttal**
> >
> > Glad to hear that and thanks for helping us improve the paper's quality.

---

### Decision · Action_Editors · 2022-07-11

**Recommendation:** Accept as is

**Comment:**

This paper considers disentangling the label information and domain style for addressing the domain generalization problem better. The general idea could lead more papers to focus on the solvability of the DG problem, which is good for the area. They also first give a theoretical bound and then design their algorithm, making the whole algorithm sound and theoretical solid. The sufficient experiments verify that the proposed method works better than existing sota methods sota experiments. In general, the main idea of this paper and the theory proposed in this paper are interesting. The authors address reviewers' concerns well in their rebuttal. Thus, I would like to recommend accepting the submission as is.